# Toward a Characterization of Loss Functions for Distribution Learning

**Nika Haghtalab**[*]
Cornell University
nika@cs.cornell.edu

**Cameron Musco**[*]
UMass Amherst
cmusco@cs.umass.edu

**Bo Waggoner**[†]
U. Colorado
bwag@colorado.edu

## Abstract

In this work we study loss functions for learning and evaluating probability distributions over large discrete domains. Unlike classification or regression where a wide variety of loss functions are used, in the distribution learning and density estimation literature, very few losses outside the dominant *log loss* are applied. We aim to understand this fact, taking an axiomatic approach to the design of loss functions for distributions. We start by proposing a set of desirable criteria that any good loss function should satisfy. Intuitively, these criteria require that the loss function faithfully evaluates a candidate distribution, both in expectation and when estimated on a few samples. Interestingly, we observe that *no loss function* possesses all of these criteria. However, one can circumvent this issue by introducing a natural restriction on the set of candidate distributions. Specifically, we require that candidates are *calibrated* with respect to the target distribution, i.e., they may contain less information than the target but otherwise do not significantly distort the truth. We show that, after restricting to this set of distributions, the log loss and a large variety of other losses satisfy the desired criteria. These results pave the way for future investigations of distribution learning that look beyond the log loss, choosing a loss function based on application or domain need.

## 1 Introduction

Estimating a probability distribution given independent samples from that distribution is a fundamental problem in machine learning and statistics [e.g. 23, 2, 24, 5]. In machine learning applications, the distribution of interest is often over a very large but finite sample space, e.g., the set of all English sentences up to a certain length or images of a fixed size in their RGB format.

A central problem is **evaluating** the learned distribution, most commonly using a loss function. Such evaluation is an important task in its own right as well as central to some learning techniques. Given a ground truth distribution $\mathbf{p}$ over a set of outcomes $\mathcal{X}$ and a sample $x \sim \mathbf{p}$, a loss function $\ell(\mathbf{q}, x)$ evaluates the performance of a candidate distribution $\mathbf{q}$ in predicting $x$. Generally, $\ell(\mathbf{q}, x)$ will be higher if $\mathbf{q}$ places smaller probability on $x$. Thus, in expectation over $x \sim \mathbf{p}$, the loss will be lower for candidate distributions that closely match $\mathbf{p}$.

The dominant loss applied in practice is the log loss $\ell(\mathbf{q}, x) = \ln(1/q_x)$, which corresponds to the learning technique of log likelihood maximization. Surprisingly, few other losses are ever considered. This is in sharp contrast to other areas of machine learning, including in supervised learning where different applications have necessitated the use of different losses, such as the squared loss, hinge loss, $\ell_1$ loss, etc. However, alternative loss functions can be beneficial for distribution learning on large domains, as we show with a brief motivating example.

---

[*]Research conducted while at Microsoft Research, New England.
[†]Research conducted while at Microsoft Research, New York City.

**Motivating example.** In many learning applications, one seeks to fit a complex distribution with a simple model that cannot fully capture its complexity. This includes e.g., noise tolerant or agnostic learning. As an example, consider modeling the distribution over English words with a character trigram model. While this model, trained by minimizing log loss, fits the distribution of English words relatively well, its performance significantly degrades if a small amount of mostly-irrelevant data is added, e.g. if the dataset includes a small fraction of foreign language words. The model is unable to fit the 'tail' of the distribution (corresponding to foreign words), however, in trying to do so it performs significantly worse on the 'head' of the distribution (corresponding to common English words). This is due to the fact that minimizing log loss requires $q_x$ to not be much smaller than $p_x$ for all $x$. A more robust loss function, such as the *log log loss*, $\ell(\mathbf{q}, x) = \ln(\ln(1/q_x))$, emphasizes the importance of fitting the 'head' and is less sensitive to the introduction of the foreign words. See Figure 1 and the full version of the paper for details.

| Samples from $\mathbf{q}_1$ | Samples from $\mathbf{q}_2$ |
|---|---|
| brappost | to |
| hild | oneems |
| me | the |
| on | not |
| ther | of |

| log loss($\mathbf{p}$) = 7.45 |
|---|
| log log loss($\mathbf{p}$) = 1.91 |
| log loss($\mathbf{q}_1$) = 11.25 |
| log log loss($\mathbf{q}_1$) = 2.22 |
| log loss($\mathbf{q}_2$) = 12.26 |
| log log loss($\mathbf{q}_2$) = 2.18 |

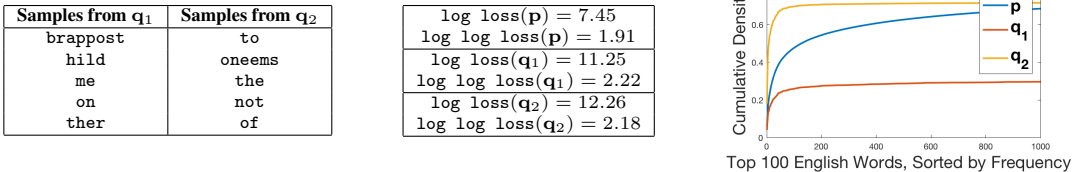

Figure 1: Modeling the distribution of English words, corrupted with 12% French and German words with character trigrams. Distribution $\mathbf{q}_1$ is trained by minimizing log loss. $\mathbf{q}_2$ achieves worse log loss but better *log log loss* and better performance at fitting the 'head' of the the target $\mathbf{p}$, indicating that log log loss may be more appropriate in this application. See the full version for more details.

**Loss function properties.** In this paper, we start by understanding the desirable properties of log loss and seek to identity other loss functions with such properties that can have applications in various domains. A key characteristic of the log loss is that it is (strictly) *proper*. That is, the true underlying distribution $\mathbf{p}$ (uniquely) minimizes the expected loss on samples drawn from $\mathbf{p}$. Properness is essential for loss functions, as without it minimizing the expected loss leads to choosing an incorrect candidate distribution even when the target distribution is fully known. Log loss is also *local* (sometimes termed *pointwise*). That is, the loss of $\mathbf{q}$ on sample $x$ is a function of the probability $q_x$ and not of $q_{x'}$ for $x' \neq x$. Local losses are preferred in machine learning, where $q_x$ is often implicitly represented as the output of a likelihood function applied to $x$, but where fully computing $\mathbf{q}$ requires at least linear time in the size of the sample space $N$ and is infeasible for large domains, such as learning the distribution of all English sentences up to a certain length.

It is well-known that *log loss is the unique local and strictly proper loss function* [19, 22, 13]. Thus, requiring strict properness and locality already restricts us to using the log loss. At the same time, these restrictive properties are not sufficient for effective distribution learning, because:

- A candidate distribution may be far from the target yet have arbitrarily close to optimal loss. Motivated by this problem, we define *strongly proper* losses that, if given a candidate far from the target, will give an expected loss significantly worse than optimal.

- A candidate distribution might be far from the target, yet on a small number of samples, it may be likely to have smaller empirical loss than that of the target. This motivates our definition of *sample-proper* losses.

- On a small number of samples, the empirical loss of a distribution may be far from its expected loss, making evaluation impossible. This motivates our definition of *concentrating* losses.

Naively, it seems we cannot satisfy all our desired criteria: our only local strictly proper loss is the log loss, which in fact fails to satisfy the concentration requirement (see Example 4). We propose to overcome this challenge by restricting the set of candidate distributions, specifically to ones that satisfy the reasonable condition of *calibration*. We then consider the properties of loss functions on, not the set of all possible distributions, but the set of calibrated distributions.

**Calibration and results.** We call a candidate distribution $\mathbf{q}$ calibrated with respect to a target $\mathbf{p}$ if all elements to which $\mathbf{q}$ assigns probability $\alpha$ actually occur on average with probability $\alpha$ in

the target distribution.[3] This can also be interpreted as requiring $\mathbf{q}$ to be a coarsening of $\mathbf{p}$, i.e., a calibrated distribution may contain less information than $\mathbf{p}$ but does otherwise not distort information. While for simplicity we focus on exactly calibrated distributions, in the full version we extend our results to a natural notion of approximate calibration. Our main results show that the calibration constraint overcomes the impossibility of satisfying properness along with the our three desired criteria.

**Main results** (Informal summary). *Any (local) loss $\ell(\mathbf{q}, x) := f\left(\frac{1}{q_x}\right)$ such that $f$ is strictly concave and monotonically increasing has the following properties subject to calibration:*

1. *$\ell$ is strictly proper, i.e., the target distribution minimizes expected loss.*

2. *If in addition $f$ satisfies left-strong-concavity, then $\ell$ is strongly proper, i.e., distributions far from the target have significantly worse loss.*

3. *If in addition to the above $f$ grows relatively slowly, then $\ell$ is sample proper i.e., on few samples, distributions far from the target have higher empirical loss with high probability.*

4. *Under these same conditions, $\ell$ concentrates i.e., on few samples, a distribution's empirical loss is a reliable estimate of its expected loss with high probability.*

The above criteria are formally introduced in Section 3. Each criteria is parameterized and different losses satisfy them with different parameters. We illustrate a few examples in Table 1 below. We emphasize that all losses shown below achieve relatively strong bounds, only depending polylogarithmically on the domain size $N$. Thus, we view all of these loss functions as viable alternatives to the log loss, which may be useful in different applications.

| $\ell(\mathbf{q}, x)$ | Strong Properness $\mathbb{E}\,\ell(\mathbf{q}; x) - \mathbb{E}\,\ell(\mathbf{p}; x)$ | Concentration sample size $m(\gamma, N)$ | Sample Properness sample size $m(\epsilon, N)$ |
|---|---|---|---|
| $\ln \frac{1}{q_x}$ | $\Omega(\epsilon^2)$ | $\tilde{O}\left(\gamma^{-2} \ln \left(\frac{N}{\gamma}\right)^2\right)$ | $O\left(\epsilon^{-4} (\ln N)^2\right)$ |
| $\left(\ln \frac{1}{q_x}\right)^p$ for $p \in (0, 1]$ | $\Omega\left(\epsilon^2 (\ln N)^{p-1}\right)$ | $\tilde{O}\left(\gamma^{-2} \ln \left(\frac{N}{\gamma}\right)^{2p}\right)$ | $O\left(\epsilon^{-4} (\ln N)^2\right)$ |
| $\ln \ln \frac{1}{q_x}$ | $\Omega\left(\frac{\epsilon^2}{\ln N}\right)$ | $\tilde{O}\left(\gamma^{-2} \ln \ln \left(\frac{N}{\gamma}\right)^2\right)$ | $O\left(\epsilon^{-4}(\ln \ln N)^2(\ln N)^2\right)$ |
| $\left(\ln \frac{e^2}{q_x}\right)^2$ | $\Omega(\epsilon^2)$ | $\tilde{O}\left(\gamma^{-2} \ln \left(\frac{N}{\gamma}\right)^4\right)$ | $O\left(\epsilon^{-4}(\ln N)^4\right)$ |

Table 1: Examples of loss function that demonstrate strong properness, sample properness, and concentration, when restricted to calibrated distributions. In the above, $N$ is the distributions support size, $\epsilon := \|\mathbf{p} - \mathbf{q}\|_1$ is the $\ell_1$ distance between $\mathbf{p}$ and $\mathbf{q}$, and $\gamma$ is an approximation parameter for concentration (see Section 4.2 for details). We assume for simplicity that $\epsilon \geq 1/N$ and hide dependencies on a success probability parameter for sample properness and concentration. $\tilde{O}(\cdot)$ suppresses logarithmic dependence on $1/\epsilon$ and $1/\gamma$.

## 1.1 Related work

Our work is directly inspired by applications of distribution estimation in very high-dimensional spaces, such as language modeling [18]. However, we do not know of work in this area that takes a systematic approach to designing loss functions.

A conceptually related research problem is that of learning distributions using computationally and statistically efficient algorithms. Beyond loss function minimization, a number of general-purpose methods have been proposed for this problem, including using histograms, nearest neighbor estimators, etc. See [15] for a survey of these methods. Much of the work in this space focuses on learning *structured* or *parametric* distributions [7, 16, 17, 6], e.g., monotone distributions or mixtures of Gaussians. On the other hand, learning an unstructured discrete distribution with support size $N$ within $\ell_1$ distance $\epsilon$ requires $\mathrm{poly}(N, 1/\epsilon)$ samples. Thus, works in this space typically focus on designing computationally efficient algorithms for optimal estimation using large sample sets [24]. In comparison, we focus on unstructured distributions with prohibitively large supports and

characterize loss functions that only require $\mathrm{polylog}(N)$ sample complexity to estimate. We do not introduce a general algorithm for distribution learning — as any such algorithm would require $\Omega(N)$ samples. Rather, motivated by tailored algorithms used in complex domains such as natural language processing, our work characterizes loss functions that could be used by a variety of algorithms.

Outside distribution learning, loss functions (termed *scoring rules*) have been studied for decades in the information elicitation literature, which seeks to incentivize experts, such as weather forecasters, to give accurate predictions [e.g. 4, 14, 22, 12, 13]. The notion of loss function properness, for example, comes from this literature. Recent research has made some connections between information elicitation and loss functions in machine learning; however, it has focused mostly on the classification and regression and not distribution learning [1, 12, 20, 21, 9]. Our work can be viewed as a contribution to the literature on evaluating forecasters by showing that, if the forecaster is constrained to be calibrated, then a variety of simple local loss functions become (strongly, sample) proper.

## 2  Preliminaries

We work with distributions over a finite domain $\mathcal{X}$ with $|\mathcal{X}| = N$. The set of all distributions over $\mathcal{X}$ is denoted by $\Delta_{\mathcal{X}}$. We denote a distribution $\mathbf{p} \in \{0, 1\}^N$ over $\mathcal{X}$ by a vector of probabilities, where $p_x$ is the probability $\mathbf{p}$ places on $x \in \mathcal{X}$. For any set $B \subseteq \mathcal{X}$, the total probability $\mathbf{p}$ places on $B$ is denoted by $\mathbf{p}(B) := \sum_{x \in B} p_x$. We use $X$ to denote a random variable on $\mathcal{X}$ whose distribution is specified in context. We also consider point mass distributions $\boldsymbol{\delta}^x \in \Delta_{\mathcal{X}}$ where $\delta_{x'}^x = \mathbf{1}\,[x = x']$.

Throughout this paper, we typically use $\mathbf{p}$ to denote the true (or target) distribution and $\mathbf{q}$ to denote a candidate or predicted distribution. For any two distributions $\mathbf{p}$ and $\mathbf{q}$, the *total variation distance* between them is defined by $\mathrm{TV}(\mathbf{p}, \mathbf{q}) := \sup_{B \subseteq \mathcal{X}} \mathbf{p}(B) - \mathbf{q}(B) = \frac{1}{2}\|\mathbf{p} - \mathbf{q}\|_1$, where $\|\cdot\|_1$ denotes the $\ell_1$ norm of a vector. Together, $\ell_1$ and the total variation distance are two of the most widely used measures of distance between distributions.

To measure the quality of a candidate distribution $\mathbf{q}$ given samples from $\mathbf{p}$, machine learning typically turns to loss functions. A *loss function* is a function $\ell : \Delta_{\mathcal{X}} \times \mathcal{X} \to \mathbb{R}$ where $\ell(\mathbf{q}, x)$ is the loss assigned to candidate $\mathbf{q}$ on outcome $x$. Given a target distribution $\mathbf{p}$, the *expected loss* for candidate $\mathbf{q}$ is defined as $\ell(\mathbf{q}; \mathbf{p}) := \mathbb{E}_{X \sim \mathbf{p}}\left[\ell(\mathbf{q}, X)\right]$. A loss function is called *proper* if $\ell(\mathbf{p}; \mathbf{p}) \leq \ell(\mathbf{q}; \mathbf{p})$ for all $\mathbf{p} \neq \mathbf{q}$, and *strictly proper* if the inequality is always strict[4]. Two common examples of proper loss functions are the *log loss* function $\ell(\mathbf{q}, x) = \ln(\frac{1}{q_x})$ (with the logarithm always taken base $e$ in this paper) and the *quadratic loss* $\ell(\mathbf{q}, x) = \frac{1}{2}\|\boldsymbol{\delta}^x - \mathbf{q}\|_2^2$. A loss function $\ell$ is called *local* if $\ell(\mathbf{q}, x)$ is a function of $q_x$ alone. For example, the log loss is local while the quadratic loss is not.

Our main results are characterized by the geometry of the loss functions we consider. For simplicity, we will generally assume functions are differentiable, although our results can be extended.

**Definition 1** (Strongly Concave). A function $f : [0, \infty] \to \mathbb{R}$ is $\beta$-*strongly concave* if for all $z, z'$ in the domain of $f$, $f(z) \leq f(z') + \nabla f(z') \cdot (z - z') - \frac{\beta}{2}(z - z')^2$.

We also consider a relaxation of strong concavity that helps us in analyzing functions that have a large curvature close to the origin but flatten out as we move farther from it.

**Definition 2** (Left-Strongly Concave). A function $f : [0, \infty] \to \mathbb{R}$ is $\beta(z)$-*left-strongly concave* if the function restricted to $[0, z]$ is $\beta(z)$-strongly concave, for all $z$.

As discussed, a natural assumption on the set of candidate distributions is *calibration*. Formally:

**Definition 3** (Calibration). Given a distribution $\mathbf{q} \in \Delta_{\mathcal{X}}$, let $B_t(\mathbf{q}) = \{x : q_x = t\}$. When it is clear from the context, we suppress $\mathbf{q}$ in the definition of $B_t$. We say that $\mathbf{q}$ is *calibrated with respect to* $\mathbf{p}$, if $\mathbf{q}(B_t(\mathbf{q})) = \mathbf{p}(B_t(\mathbf{q}))$ for all $t \in [0, 1]$. We let $\mathcal{C}(\mathbf{p})$ denote the set of all calibrated distributions with respect to $\mathbf{p}$.

In other words, $\mathbf{q}$ is calibrated with respect to $\mathbf{p}$ if points assigned probability $q_x = t$ have average probability $t$ under $\mathbf{p}$. In other words, $\mathbf{p}$ can be "coarsened" to $\mathbf{q}$ by taking subsets of points and replacing their probabilities with the subset average. Note that the uniform distribution $\mathbf{q} = \left(\frac{1}{N}, \dots, \frac{1}{N}\right)$ is calibrated with respect to all $\mathbf{p}$, and that $\mathbf{p}$ is calibrated with respect to itself. Also

note that there are only finitely many values $t \in [0, 1]$ for which $B_t$ is non-empty. We denote the set of these values by $T(\mathbf{q}) = \{t : B_t \neq \emptyset\}$.

We refer an interested reader to the full version of the paper for a more detailed discussion of the notion of calibration and its connections to similar notions used in forecasting theory, e.g. [8, 11]. See the full version for a discussion of how our results can be extended to a natural notion of approximate calibration.

# 3   Three Desirable Properties of Loss Functions

In this section, we define three criteria and discuss why any desirable loss function should demonstrate them. We use examples of loss functions, such as the log loss $\ell_{log\text{-}loss}(\mathbf{q}, x) = \ln(\frac{1}{q_x})$ and the linear loss $\ell_{lin\text{-}loss}(\mathbf{q}, x) = -q_x$ to help demonstrate the existence or lack of these criteria.

## 3.1   Strong Properness

Recall that a loss function is strictly proper if all incorrect candidate distributions yield a higher expected loss value than the target distribution. Here, we expand this to *strong* properness where this gap in expected loss grows with distance from the target distribution. We also extend both definitions to hold over a specific domain of candidate distributions, rather than all distributions.

**Definition 4** (Calibrated Properness). Let $\mathcal{P} : \Delta_{\mathcal{X}} \to 2^{\Delta_{\mathcal{X}}}$ be a *domain function*, that is, $\mathcal{P}(\mathbf{p}) \subseteq \Delta_{\mathcal{X}}$ is a restricted set of distributions. A loss function $\ell$ is *proper over* $\mathcal{P}$ if for all $\mathbf{p} \in \Delta_{\mathcal{X}}$, $\mathbf{p} \in \operatorname{argmin}_{\mathbf{q} \in \mathcal{P}(\mathbf{p})} \ell(\mathbf{q}; \mathbf{p})$. A loss function is said to be *strictly proper over* $\mathcal{P}$ if the argmin is always unique. When $\mathcal{P}(\mathbf{p}) = \mathcal{C}(\mathbf{p})$, i.e. is the set of calibrated distributions w.r.t. $\mathbf{p}$, we call such a loss function *(strictly) calibrated proper*.

**Example 1.** It is well-known that $\ell_{log\text{-}loss}(\mathbf{q}, x) = \ln\left(\frac{1}{q_x}\right)$ is the *unique local* proper loss function (up to scaling) over the unrestricted domain $\mathcal{P}(\mathbf{p}) = \Delta_{\mathcal{X}}$ [3]. Indeed, it is known that the difference in expected log loss of a prediction $\mathbf{q}$ and the target distribution $\mathbf{p}$ is the KL-divergence, i.e.

$$\ell_{log\text{-}loss}(\mathbf{q}; \mathbf{p}) - \ell_{log\text{-}loss}(\mathbf{p}; \mathbf{p}) = \mathrm{KL}(\mathbf{p}, \mathbf{q}) := \sum_x p_x \ln\left(\frac{p_x}{q_x}\right). \tag{1}$$

Furthermore, the KL-divergence is strictly positive for $\mathbf{p} \neq \mathbf{q}$. This proves that the log loss is strictly proper over $\Delta_{\mathcal{X}}$, and as a result, is strictly calibrated proper as well.

On the other hand, $\ell_{lin\text{-}loss}(\mathbf{q}, x) = -q_x$ is not proper over $\Delta_{\mathcal{X}}$. This is due to that fact that the minimizer of this loss is the point mass distribution $\boldsymbol{\delta}^x$ for $x = \operatorname{argmax}_x p_x$. For example, for target distribution $\mathbf{p} = (\frac{1}{3}, \frac{2}{3})$, distribution $\mathbf{q} = (0, 1)$ yields a lower $\ell_{lin\text{-}loss}$ than that of $\mathbf{p}$. Note, however, that such a choice of $\mathbf{q}$ is not calibrated with respect to $\mathbf{p}$. When loss minimization is constrained to the set of calibrated distributions, $\mathcal{C}(\mathbf{p}) = \{(\frac{1}{3}, \frac{2}{3}), (\frac{1}{2}, \frac{1}{2})\}$, $\mathbf{p}$ minimizes the expected linear loss. Indeed, in Section 4 we show more generally that the linear loss and in fact many reasonable local loss functions are calibrated proper.

While strict properness is an important baseline guarantee, we would like a "stronger" property: If $\mathbf{q}$ is significantly incorrect in the sense of being far from $\mathbf{p}$, then the expected loss of $\mathbf{q}$ should be significantly worse. This motivates the following definition. An analogous definition has appeared in the context of mechanism design in [10].

**Definition 5** (Strong Calibrated Properness). A loss function $\ell$ is *$\beta$-strongly proper over a domain function* $\mathcal{P}$ if for all $\mathbf{p} \in \Delta_{\mathcal{X}}$, for all $\mathbf{q} \in \mathcal{P}(\mathbf{p})$, $\ell(\mathbf{q}; \mathbf{p}) - \ell(\mathbf{p}; \mathbf{p}) \geq \frac{\beta}{2} \|\mathbf{p} - \mathbf{q}\|_1^2$. When $\mathcal{P}(\mathbf{p}) = \mathcal{C}(\mathbf{p})$, we call such functions *$\beta$-strongly calibrated proper* and when $\mathcal{P}(\mathbf{p}) = \Delta_{\mathcal{X}}$, we simply refer to them as *$\beta$-strongly proper*.

**Example 2.** The log loss is $1$-strongly proper. This is *equivalent* to Pinsker's inequality, which states that *for all $\mathbf{p}$ and $\mathbf{q}$*, $\mathrm{KL}(\mathbf{p}, \mathbf{q}) \geq 2\mathrm{TV}(\mathbf{p}, \mathbf{q})^2$. Together with (1) and the fact that $\mathrm{TV}(\mathbf{p}, \mathbf{q}) = \frac{1}{2}\|\mathbf{p} - \mathbf{q}\|_1$, this shows that log loss is $1$-strongly proper (and thus also $1$-strongly calibrated proper.)

As we will see in Section 4, strong calibrated properness relates to the notion of strong concavity (of the inverse loss function) in $\ell_1$ norm. We refer the interested reader to the full version of the paper for a discussion of the use of alternative norms in the definition of strong properness. In the full version

we extend the study of normed concavity of loss functions to strong properness of a loss function over $\Delta_{\mathcal{X}}$ .

## 3.2 Sample-properness

So far, we have focused on the loss a candidate $\mathbf{q}$ receives in *expectation over $x \sim \mathbf{p}$*. Of course, if one is attempting to learn $\mathbf{p}$, this expectation can generally not be computed. We would like the notion of properness to carry over to the setting when the loss on $\mathbf{q}$ is estimated using a small set of samples from $\mathbf{p}$. We say that a loss function is sample-proper if within a small number, all candidate distributions that are sufficiently far from $\mathbf{p}$ yield a loss that is larger than that of $\mathbf{p}$ on the samples.

In the remainder of this paper, let $\hat{\mathbf{p}}$ denote the empirical distribution corresponding to samples drawn from $\mathbf{p}$. Note that the average loss of any $\mathbf{q}$ on the samples can be written $\ell(\mathbf{q}; \hat{\mathbf{p}})$. Formally:

**Definition 6** (Calibrated Sample-Properness). A loss function $\ell$ is $m(\epsilon, \delta, N)$-*sample proper* over a function domain $\mathcal{P}$ if, for all $\mathbf{p} \in \Delta_{\mathcal{X}}$ and all $\mathbf{q} \in \mathcal{P}(\mathbf{p})$ with $\|\mathbf{p} - \mathbf{q}\|_1 \geq \epsilon$, with probability at least $1 - \delta$ over $m(\epsilon, \delta, N)$ i.i.d. samples from $\mathbf{p}$, we have $\ell(\mathbf{p}; \hat{\mathbf{p}}) < \ell(\mathbf{q}; \hat{\mathbf{p}})$. When $\mathcal{P}(\mathbf{p}) = \mathcal{C}(\mathbf{p})$, we call such functions *calibrated $m(\epsilon, \delta, N)$-sample proper.*

**Example 3.** A folklore theorem states that $\ell_{log\text{-}loss}$ is $O\left(\frac{1}{\epsilon^2} \ln\left(\frac{1}{\delta}\right)\right)$-sample proper over $\Delta_{\mathcal{X}}$, and as a result it is calibrated $O\left(\frac{1}{\epsilon^2} \ln\left(\frac{1}{\delta}\right)\right)$-sample proper.

Now consider $\ell_{lin\text{-}loss}(\mathbf{q}, x) = -q_x$. Since it is not a proper loss function over $\Delta_{\mathcal{X}}$, by definition it is not sample proper over $\Delta_{\mathcal{X}}$ for any $m(\epsilon, \delta, N)$. When restricting to calibrated distributions however, as we claimed in Example 1 linear loss is calibrated proper in expectation. It is interesting to note that linear loss is not sample proper for any $m(\epsilon, \delta, N) \in o\left(N\right)$. To observe this, consider $\mathbf{p}$ where $p_1 = \frac{1}{4} + \frac{1}{\sqrt{m}}$, $p_2 = \frac{1}{4} - \frac{1}{\sqrt{m}}$, and $p_x = \frac{1}{2(N/2-2)}$ for $x = 3, \ldots, N/2$ and $p_x = 0$ for $x = N/2 + 1, \ldots, N$. Consider $\mathbf{q}$ where $q_1 = q_2 = \frac{1}{4}$ and $q_x = \frac{1}{2(N-2)}$ for $x = 3, \ldots, N$. Let $\hat{\mathbf{p}}$ be the empirical distribution. With a constant probability, $\hat{p}_1 \leq \frac{1}{4} - \frac{1}{\sqrt{m}}$ and $\hat{p}_2 \geq \frac{1}{4}$. Let $\nu = \frac{1}{2(N/2-2)} - \frac{1}{2(N-2)} = \Theta(\frac{1}{N})$. Therefore,

$$\ell(\mathbf{q}; \hat{\mathbf{p}}) - \ell(\mathbf{p}; \hat{\mathbf{p}}) = \sum_{x=1}^{N} \hat{p}_x (p_x - q_x)$$

$$= \frac{\hat{p}_1}{\sqrt{m}} + \frac{-\hat{p}_2}{\sqrt{m}} + \nu \sum_{x=3}^{N/2} \hat{p}_x - \nu \sum_{x=N/2+1}^{N} \hat{p}_x$$

$$= \frac{1}{\sqrt{m}} \left( \frac{1}{4} - \frac{1}{\sqrt{m}} \right) - \frac{1}{\sqrt{m}} \frac{1}{4} + \Theta\left( \frac{1}{N} \right)$$

$$= -\frac{1}{m} + \Theta\left( \frac{1}{N} \right) < 0,$$

when $m \in o\left(N\right)$. Furthermore, note that $\mathbf{q}$ is calibrated w.r.t. $\mathbf{p}$ with two non-empty buckets $B_{\frac{1}{4}}(\mathbf{q}) = \{1, 2\}$ and $B_{\frac{1}{2(N-2)}}(\mathbf{q}) = \{3, \ldots, N\}$. Moreover, $\|\mathbf{p} - \mathbf{q}\|_1 = \Theta(1)$. Thus, for $\ell_{lin\text{-}loss}$ to be calibrated $m(\epsilon, \delta, N)$-sample proper, we must have $m(\Theta(1), \Theta(1), N) \in \Omega\left(N\right)$.

## 3.3 Concentration

Beyond sample properness, when the expected loss $\ell(\mathbf{q}; \mathbf{p})$ is estimated from a small i.i.d. sample from $\mathbf{p}$, we would like the empirical loss to remain faithful to the true value. For example, one might hope that minimizing loss on that sample will result in a distribution that has small loss on $\mathbf{p}$. This will hold as long as the empirical loss well approximates the true expected loss with high probability.

**Definition 7** (Calibrated Concentration). A loss function $\ell$ *concentrates over domain function $\mathcal{P}$ with $m(\gamma, \delta, N)$ samples* if for all $\mathbf{p} \in \Delta_{\mathcal{X}}$, for all $\mathbf{q} \in \mathcal{P}(\mathbf{p})$, for $m(\gamma, \delta, N)$ i.i.d. samples from $\mathbf{p}$, $\Pr\left[ |\ell(\mathbf{q}; \hat{\mathbf{p}}) - \ell(\mathbf{q}; \mathbf{p})| \geq \gamma \right] \leq \delta$. When $\mathcal{P}(\mathbf{p}) = \mathcal{C}(\mathbf{p})$, we say that $\ell$ *calibrated concentrates* with $m(\gamma, \delta, N)$ samples.[5]

**Example 4.** We can easily see that log loss does *not* concentrate with $o(N)$ samples over $\Delta_\mathcal{X}$. Let $\mathbf{p}$ be the uniform distribution and $\mathbf{q}$ be uniform on $\mathcal{X} \setminus \{x\}$. With high probability, $x$ is not sampled, and $\ell(\mathbf{q}; \hat{\mathbf{p}})$ is finite. Yet $\ell(\mathbf{q}; \mathbf{p}) = \infty$. Note that although this example is extreme, its conclusion is robust: one can make an arbitrarily large finite gap. As we will see, the log loss, along with many other reasonable loss will concentrate with a small number of samples over calibrated distributions.

## 4 Main Results

Looking back at the criteria defined in Section 3, we are immediately faced with an impossibility result: no local loss function exists that satisfies properness, $o(N)$-sample properness, and concentration with $o(N)$ samples. This is because log loss is the unique local loss function that satisfies the first property and as shown in Example 4 it does not concentrate. In this section, we show that a broad class of local loss functions with certain niceness properties satisfies the above three criteria over calibrated domains. Specifically, we consider loss functions $\ell(\mathbf{q}, x)$ that are non-increasing in $q_x$ and are inversely concave: $\ell(\mathbf{q}, x) = f(\frac{1}{q_x})$ for some concave function $f$. Similarly, we say that $\ell$ is inversely strongly concave if the corresponding $f$ is strongly concave.

### 4.1 Calibrated and Strong Calibrated Properness

In this section, we show that any (strongly) nice loss function is (strongly) proper over the domain of calibrated distributions. More formally.

**Theorem 1** (Strict Properness). *Suppose the local loss function $\ell$ is such that $\ell(\mathbf{q}, x) = f(\frac{1}{q_x})$ for a concave $f$ function. Then, $\ell$ is strictly proper over the domain function $\mathcal{C}$.*

**Theorem 2** (Strong Properness). *Suppose the loss function $\ell$ is such that $\ell(\mathbf{q}, x) = f(\frac{1}{q_x})$ where $f$ is non-decreasing and is $\frac{C(x)}{x^2}$-left-strongly concave where $C(x)$ is non-increasing and non-negative for $x \geq 1$. Then for all $\mathbf{p} \in \Delta_\mathcal{X}$ and $\mathbf{q} \in \mathcal{C}(\mathbf{p})$,*

$$\ell(\mathbf{q}; \mathbf{p}) - \ell(\mathbf{p}; \mathbf{p}) \geq C\left(\frac{4N}{\|\mathbf{p} - \mathbf{q}\|_1}\right) \cdot \frac{\|\mathbf{p} - \mathbf{q}\|_1^2}{128}.$$

We defer the proof of Theorem 2 to the full version.

We begin with the proof of Theorem 1, which relies on a key property of calibration stated in Lemma 1. At a high level, this lemma shows that the average value of $1/p_x$ and $1/q_x$ is the same over instances $x$ such that $q_x = t$, which is also equal to $1/t$.

**Lemma 1.** *For any distribution $\mathbf{p} \in \Delta_\mathcal{X}$ and $\mathbf{q} \in \mathcal{C}(\mathbf{p})$, and for any $t \in [0, 1]$, we have $\mathbb{E}_{X \sim \mathbf{p}}\left[\frac{1}{p_X} \mid X \in B_t\right] = \frac{1}{t}$, where $B_t = \{x : q_x = t\}$.*

*Proof.* We have

$$\mathbb{E}\left[\frac{1}{p_X} \mid X \in B_t\right] = \sum_{x \in B_t} \frac{p_x}{\mathbf{p}(B)} \frac{1}{p_x} = \frac{|B_t|}{\mathbf{p}(B_t)} = \frac{1}{t}.$$

$\square$

*Proof of Theorem 1.* Suppose $\ell(\mathbf{q}, x) = f(\frac{1}{q_x})$ for a strictly concave $f$. Consider any $\mathbf{q}$ that is calibrated with respect to $\mathbf{p}$. Recall that $B_t = \{x : q_x = t\}$ and $T(\mathbf{q}) = \{t : |B_t| \neq \emptyset\}$ is a finite set.

$$\ell(\mathbf{p}; \mathbf{p}) = \sum_{t \in T(\mathbf{q})} \mathbf{p}(B_t) \mathbb{E}\left[f\left(\frac{1}{p_X}\right) \mid X \in B_t\right] \leq \sum_{t \in T(\mathbf{q})} \mathbf{p}(B_t) f\left(\mathbb{E}\left[\frac{1}{p_X} \mid X \in B_t\right]\right)$$

$$= \sum_{t \in T(\mathbf{q})} \mathbf{p}(B_t) f\left(\frac{1}{t}\right) = \sum_{t \in T(\mathbf{q})} \sum_{x \in B_t} p_x f\left(\frac{1}{q_x}\right) = \ell(\mathbf{q}; \mathbf{p}),$$

where the second transition is by Jensen's inequality and the third transition is by Lemma 1. If $f$ is strictly concave and there exists a $B_t$ where $\mathbf{q}$ and $\mathbf{p}$ disagree, then the inequality is strict. $\square$

## 4.2 Concentration

The (strong) properness of a loss function, as discussed in Section 4.1, is only concerned with loss functions in expectation. In this section, we consider finite sample guarantees. Recall that $\ell$ concentrates over $\mathcal{P}(\mathbf{p})$ (Definition 7) if, with $m(\gamma, \delta, N)$ samples, the empirical loss $\ell(\mathbf{q}; \hat{\mathbf{p}})$ of a distribution $\mathbf{q} \in \mathcal{P}(\mathbf{p})$ is $\gamma$-close to its true loss $\ell(\mathbf{q}; \mathbf{p})$ with probability $1 - \delta$. Concentration can be difficult to achieve: by Example 4, even the log loss does not concentrate for any sample size $o(N)$ for general $\mathbf{q} \in \Delta_{\mathcal{X}}$. However, as we show below, when $\mathbf{q}$ is *calibrated*, many natural loss functions, including log loss, indeed concentrate. All that is needed is that the loss function is inverse concave, increasing, and does not grow too quickly as $q_x \to 0$.

**Theorem 3** (Concentration). *Suppose $\ell$ is a local loss function with $\ell(\mathbf{q}, x) = f\left(\frac{1}{q_x}\right)$ for nonnegative, increasing, concave $f(z)$. Suppose further that $f(z) \leq c\sqrt{z}$ for all $z \geq 1$ and some constant $c$. Then $\ell$ concentrates over the domain function $\mathcal{C}$ for any $m(\gamma, \delta, N) \leq N$, such that*

$$m(\gamma, \delta, N) \geq \frac{c_1 \cdot f(\beta)^2 \ln \frac{1}{\delta}}{\gamma^2},$$

*where $c_1$ is a fixed constant and $\beta := \frac{16 N^8}{\delta \cdot \min(1, \gamma^2/c^2)}$. That is, for any $\mathbf{p} \in \Delta_{\mathcal{X}}, \mathbf{q} \in \mathcal{C}(\mathbf{p})$, drawing at least $m(\gamma, \delta, N)$ samples guarantees $|\ell(\mathbf{q}; \hat{\mathbf{p}}) - \ell(\mathbf{q}; \mathbf{p})| \leq \gamma$ with probability $\geq 1 - \delta$.*

Note that $\gamma$ bounds the absolute difference between $\ell(\mathbf{q}; \hat{\mathbf{p}})$ and $\ell(\mathbf{q}; \mathbf{p})$. The desired difference may depend on the relative scale of the loss function. If e.g., we take $\ell(\mathbf{q}, x)$ and scale to obtain $\ell'(\mathbf{q}, x) = \alpha \cdot \ell(\mathbf{q}, x)$ for some $\alpha$, the desired error $\gamma$ scales by $\alpha$, $f(\beta)$ and $c$ both scale by $\alpha$, and thus we can see that the sample complexity remains fixed.

We defer the proof of Theorem 3 to the full version of the paper. At a high level, Theorem 3 holds because calibration helps us avoid worst-case instances (as in Example 4) using a very simple fact: when $\mathbf{q}$ is calibrated, we have $\frac{q_x}{p_x} \geq \frac{1}{N}$ for all $x$. This rules out very low probability events that contribute significantly to $\ell(\mathbf{q}; \mathbf{p})$ but require many samples to identify. To prove Theorem 3 we partition $\mathcal{X}$ into $\Omega$ containing elements of very small probability, and $\mathcal{X} \setminus \Omega$. With high probability, no element of $\Omega$ is ever sampled from $\mathbf{p}$. Conditioned on this, the loss is bounded (and its expectation does not change much), so a concentration result can be applied.

## 4.3 Sample Properness

Lastly, we turn our attention to calibrated sample properness. Recall that a loss function is sample proper if all candidate distributions that are sufficiently far from $\mathbf{p}$ have a loss that is larger $\mathbf{p}$ on the empirical distribution $\hat{\mathbf{p}}$ corresponding to a small number of samples from $\mathbf{p}$. It is not hard to see that sample properness of a loss function is a direct consequence of its concentration and strong properness. For any candidate distribution $\mathbf{q}$ for which $\|\mathbf{q} - \mathbf{p}\|_1$ is large, strong properness (Theorem 2) implies that $\ell(\mathbf{q}; \mathbf{p})$ is significantly larger than $\ell(\mathbf{p}; \mathbf{p})$. Furthermore, concentration (Theorem 3) implies that with high probability $\ell(\mathbf{q}; \mathbf{p}) \approx \ell(\mathbf{q}; \hat{\mathbf{p}})$ and $\ell(\mathbf{p}; \mathbf{p}) \approx \ell(\mathbf{p}; \hat{\mathbf{p}})$. Therefore, with high probability, $\ell(\mathbf{q}; \hat{\mathbf{p}}) > \ell(\mathbf{p}; \hat{\mathbf{p}})$. Formally in the full version of the paper we prove:

**Theorem 4** (Sample properness). *Suppose $\ell$ is a local loss function with $\ell(\mathbf{q}, x) = f(\frac{1}{q_x})$ for nonnegative, increasing, concave $f(z)$. Suppose further that $f(z) \leq c\sqrt{z}$ for all $z \geq 1$ and some constant $c$ and that $f$ is $\frac{C(x)}{x^2}$-left-strongly concave for where $C(x)$ is nonincreasing and nonnegative for $x \geq 1$. Then for all $\mathbf{p} \in \Delta_{\mathcal{X}}$ and $\mathbf{q} \in \mathcal{C}(\mathbf{p})$, if $\hat{\mathbf{p}}$ is the empirical distribution constructed from $m$ independent samples of $\mathbf{p}$ with $m \leq N$ and*

$$m \geq \frac{c_1 \cdot f(\beta)^2 \ln \frac{1}{\delta}}{\left(C\left(\frac{4N}{\|\mathbf{p}-\mathbf{q}\|_1}\right) \|\mathbf{p} - \mathbf{q}\|^2\right)^2},$$

*where $c_1$ is constant and $\beta := \frac{288 N^8}{\delta \cdot \min\left(1, \left[C\left(\frac{4N}{\|\mathbf{p}-\mathbf{q}\|_1}\right) \frac{\|\mathbf{p}-\mathbf{q}\|_1^2}{128c}\right]^2\right)}$, then $\ell(\mathbf{q}; \hat{\mathbf{p}}) > \ell(\mathbf{p}; \hat{\mathbf{p}})$ with prob. $\geq 1 - \delta$.*

## 4.4 Application of the Main Results to Loss Functions

We now instantiate Theorems 2, 3, and 4 for one example of a natural loss function $\ell(\mathbf{q}, x) = \ln \ln(\frac{1}{q_x})$. Refer to Table 1 for other loss functions and see the full version for details on its derivation.

First, note that $\ln\ln(z)$ is $C(z)/z^2$-left-strongly concave for $C(z) = \frac{(1+\ln(z))}{\ln(z)^2}$.[6] Moreover, $C(z)$ is non-increasing and non-negative for $z \geq 1$ and $\ln\ln(z) \leq \sqrt{z}$. Using these, for any $\mathbf{p}$ and $\mathbf{q} \in \mathcal{C}(\mathbf{p})$ such that $\|\mathbf{p} - \mathbf{q}\|_1 \geq \epsilon$ we have

- By Theorem 2, $\ell(\mathbf{q}; \mathbf{p}) - \ell(\mathbf{p}; \mathbf{p}) \geq \Omega\left(\frac{\epsilon^2}{\ln(N/\epsilon)}\right)$.

- By Theorem 3, an empirical distribution $\hat{\mathbf{p}}$ of $\tilde{O}\left(\gamma^{-2}\ln\ln(N)^2\ln(1/\delta)\right)$ i.i.d samples from $\mathbf{p}$ is sufficient such that $|\ell(\mathbf{q}; \hat{\mathbf{p}}) - \ell(\mathbf{q}; \mathbf{p})| \leq \gamma$ with probability $1 - \delta$.

- By Theorem 4, an empirical distribution $\hat{\mathbf{p}}$ of $\tilde{O}\left(\epsilon^{-4}\ln\ln(N\ln(N))^2\ln(1/\delta)\ln(N)\right)$ i.i.d samples from $\mathbf{p}$ is sufficient such that $\ell(\mathbf{q}; \hat{\mathbf{p}}) > \ell(\mathbf{p}; \hat{\mathbf{p}})$ with probability $1 - \delta$.

# 5 Discussion

In this work, we characterized loss functions that meet three desirable properties: properness in expectation, concentration, and sample properness. We demonstrated that no local loss function meets all of these properties over the domain of all candidate distributions. But, if one enforces the criterion of *calibration* (or approximate calibration as discussed in the full version), then many simple loss functions have good properties for evaluating learned distributions over large discrete domains. We hope that our work provides a starting point for several future research directions.

One natural question is to understand how to select a loss function based on the application domain. Our example for language modeling, from the introduction, motivates the idea that log loss is not the best choice always. Understanding this more formally, for example in the framework of robust distribution learning, could provide a systematic approach for selecting loss functions based on the needs of the domain. Our work also leaves open the question of designing compuationally and statistically efficient learning algorithms for different loss functions under the constraint that the candidate $\mathbf{q}$ is (approximately) calibrated. One challenge in designing computationally efficient algorithms is that the space of calibrated distributions is not convex. We present some advances towards dealing with this challenge in the full version by providing an efficient procedure for 'projecting' a non-calibrated distribution on the space of approximately calibrated distribution. It remains to be seen if iteratively applying this procedure could be useful in designing an efficient algorithm for minimizing the loss on calibrated distributions.

**Acknowledgements**

We thank Adam Kalai for significant involvement in early stages of this project and for suggesting the idea of exploring alternatives to the log loss under calibration restrictions. We also thank Gautam Kamath for helpful discussions.

## Footnotes

[3]This definition is an adaptation of the standard calibration criterion applied to sequences of predictions made by a forecaster [8, 11]. See discussion in the full version of the paper.

[4]Our use of "properness" is inspired the literature on *proper scoring rules*. It is not to be confused with "properness" in learning theory where the learned hypothesis must belong to a pre-determined class of hypotheses.

[5]We use $\gamma$ to denote difference in loss to avoid confusion with $\epsilon$, which generally means a distance between distributions.

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
