[Supplementary Material · submit-full-version.pdf]

# Toward a Characterization of Loss Functions for Distribution Learning (Full Version with Appendices)

## Abstract

In this work we study loss functions for learning and evaluating probability distributions over large discrete domains. Unlike classification or regression where a wide variety of loss functions are used, in the distribution learning and density estimation literature, very few losses outside the dominant *log loss* are applied. We aim to understand this fact, taking an axiomatic approach to the design of loss functions for learning distributions. We start by proposing a set of desirable criteria that any good loss function should satisfy. Intuitively, these criteria require that the loss function faithfully evaluates a candidate distribution, both in expectation and when estimated on a few samples. Interestingly, we observe that *no loss function* possesses all of these criteria. However, one can circumvent this issue by introducing a natural restriction on the set of candidate distributions. Specifically, we require that candidates are *calibrated* with respect to the target distribution, i.e., they may contain less information than the target but otherwise do not significantly distort the truth. We show that, after restricting to this set of distributions, the log loss, along with a large variety of other losses satisfy the desired criteria. These results pave the way for future investigations of distribution learning that look beyond the log loss, choosing a loss function based on application or domain need.

## 1 Introduction

Estimating a probability distribution given independent samples from that distribution is a fundamental problem in machine learning and statistics [e.g. 25, 5, 26, 8]. In machine learning applications, the distribution of interest is often over a very large but finite sample space, e.g., the set of all English sentences up to a certain length or images of a fixed size in their RGB format.

A central technique in learning these types of distributions, encompassing, e.g., log likelihood maximization, is evaluation via a *loss function*. Given a

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

 $x_1, \ldots, x_m$ be the samples drawn from $\mathbf{p}$. Then, with a constant probability $m(\frac{1}{4} + \frac{1}{2\sqrt{m}}) \pm \sqrt{m}$ and $m(\frac{1}{4} - \frac{1}{2\sqrt{m}}) \pm \sqrt{m}$ number of these samples are on instances $x = 1$ and $x = 2$, respectively. Therefore, with a constant probability

$$\ell(\mathbf{q}; \hat{\mathbf{p}}) - \ell(\mathbf{p}; \hat{\mathbf{p}}) = \frac{1}{m} \sum_{i=1}^{m} (p_i - q_i) \leq \sqrt{m} \pm 2\sqrt{m} - \Theta\left(\frac{1}{N}\right) < 0,$$

when $m \in o\left(N^2\right)$. Furthermore, note that $\mathbf{q}$ is calibrated w.r.t. $\mathbf{p}$ and $\|\mathbf{p} - \mathbf{q}\|_1 = \Theta(1)$. Thus, for $\ell_{lin\text{-}loss}$ to be calibrated $m(\epsilon, \delta, N)$-sample proper, we must have $m(\Theta(1), \Theta(1), N) \in \Omega\left(N^2\right)$.

## 3.3 Concentration

Beyond sample properness, when the expected loss $\ell(\mathbf{q}; \mathbf{p})$ is estimated from a small i.i.d. sample from $\mathbf{p}$, we would like the empirical loss to remain faithful to the true value. For example, one might hope that minimizing loss on that sample will result in a distribution that has small loss on $\mathbf{p}$. This will hold as long as the empirical loss well approximates the true expected loss with high probability.

**Definition 7** (Calibrated Concentration). A loss function $\ell$ *concentrates over domain function $\mathcal{P}$ with $m(\gamma, \delta, N)$ samples* if for all $\mathbf{p} \in \Delta_{\mathcal{X}}$, for all $\mathbf{q} \in \mathcal{P}(\mathbf{p})$, for $m(\gamma, \delta, N)$ i.i.d. samples from $\mathbf{p}$, $\Pr\left[|\ell(\mathbf{q}; \hat{\mathbf{p}}) - \ell(\mathbf{q}; \mathbf{p})| \geq \gamma\right] \leq \delta$. When $\mathcal{P}(\mathbf{p}) = \mathcal{C}(\mathbf{p})$, we say that $\ell$ *calibrated concentrates* with $m(\gamma, \delta, N)$ samples.[3]

**Example 4.** We can easily see that log loss does *not* concentrate with $o(N)$ samples over $\Delta_{\mathcal{X}}$. Let $\mathbf{p}$ be the uniform distribution and $\mathbf{q}$ be uniform on $\mathcal{X} \setminus \{x\}$. With high probability, $x$ is not sampled, and $\ell(\mathbf{q}; \hat{\mathbf{p}})$ is finite. Yet $\ell(\mathbf{q}; \mathbf{p}) = \infty$. Note that although this example is extreme, its conclusion is robust: one can make an arbitrarily large finite gap. As we will see, the log loss, along with many other reasonable loss will concentrate with a small number of samples over calibrated distributions.

# 4 The Main Results

Looking back at the criteria defined in Section 3, we are immediately faced with an impossibility result: no local loss function exists that satisfies properness, $o(N)$-sample properness, and concentration with $o(N)$ samples. This is because log loss is the unique local loss function that satisfies the first property and as shown in Example 4 it does not concentrate. In this section, we show that a broad class of local loss functions with certain niceness properties satisfies the above three criteria over calibrated domains. Specifically, we consider loss functions $\ell(\mathbf{q}, x)$ that are non-increasing in $q_x$ and are inversely concave: $\ell(\mathbf{q}, x) = f(\frac{1}{q_x})$ for some concave function $f$. Similarly, we say that $\ell$ is inversely strongly concave if the corresponding $f$ is strongly concave.

## 4.1 Calibrated and Strong Calibrated Properness

In this section, we show that any (strongly) nice loss function is (strongly) proper over the domain of calibrated distributions. More formally.

**Theorem 1** (Strict Properness). *Suppose the local loss function $\ell$ is such that $\ell(\mathbf{q}, x) = f(\frac{1}{q_x})$ for a concave $f$ function. Then, $\ell$ is strictly proper over the domain function $\mathcal{C}$.*

**Theorem 2** (Strong Properness). *Suppose the loss function $\ell$ is such that $\ell(\mathbf{q}, x) = f(\frac{1}{q_x})$ where $f$ is non-decreasing and is $\frac{C(x)}{x^2}$-left-strongly concave where $C(x)$ is non-increasing and non-negative for $x \geq 1$. Then for all $\mathbf{p} \in \Delta_{\mathcal{X}}$ and $\mathbf{q} \in \mathcal{C}(\mathbf{p})$,*

$$\ell(\mathbf{q}; \mathbf{p}) - \ell(\mathbf{p}; \mathbf{p}) \geq C\left(\frac{4N}{\|\mathbf{p} - \mathbf{q}\|_1}\right) \cdot \frac{\|\mathbf{p} - \mathbf{q}\|_1^2}{128}.$$

We defer the proof of Theorem 2 to Appendix B.1 and only prove Theorem 1 here. To help us with this proof, let us first understand an a key property of calibration in the next lemma, whose proof appears in Appendix A.1.

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

[5]When $B = B_t(\mathbf{q})$ for some $t \in [0, 1]$, $t(B) = t$.

[6]Note that this this is different that the usual definition of $B_t = \{x : q_x = t\}$, but it is still within the same spirit of bucketing the elements based on their $q_x$ values.

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

## A  Additional Proofs for Strict Proper Losses

### A.1  Proof of Lemma 1

**Lemma 1 (restated).** *For any distribution $\mathbf{p} \in \Delta_{\mathcal{X}}$ and $\mathbf{q} \in \mathcal{C}(\mathbf{p})$, and for any $t \in [0, 1]$, we have* $\mathbb{E}_{X \sim \mathbf{p}}\left[\frac{1}{p_X} \mid X \in B_t\right] = \frac{1}{t}$, *where $B_t = \{x : q_x = t\}$.*

*Proof.* We have

$$\mathbb{E}\left[\frac{1}{p_X} \mid X \in B_t\right] = \sum_{x \in B_t} \frac{p_x}{\mathbf{p}(B)} \frac{1}{p_x} = \frac{|B_t|}{\mathbf{p}(B_t)} = \frac{1}{t}.$$

$\square$

### A.2  Proof of Theorem 1

Suppose $\ell(\mathbf{q}, x) = f(\frac{1}{q_x})$ for a strictly concave $f$. Consider any $\mathbf{q}$ that is calibrated with respect to $\mathbf{p}$. Recall that $B_t = \{x : q_x = t\}$ and $T(\mathbf{q}) = \{t : |B_t| \neq \emptyset\}$ is a finite set.

$$\begin{aligned}
\ell(\mathbf{p}; \mathbf{p}) &= \sum_{t \in T(\mathbf{q})} \mathbf{p}(B_t) \, \mathbb{E}\left[\ell(\mathbf{p}, X) \mid X \in B_t\right] \\
&= \sum_{t \in T(\mathbf{q})} \mathbf{p}(B_t) \, \mathbb{E}\left[f\left(\frac{1}{p_X}\right) \mid X \in B_t\right] \\
&\leq \sum_{t \in T(\mathbf{q})} \mathbf{p}(B_t) f\left(\mathbb{E}\left[\frac{1}{p_X} \mid X \in B_t\right]\right) \quad \text{(By Jensen's inequality)} \\
&= \sum_{t \in T(\mathbf{q})} \mathbf{p}(B_t) f\left(\frac{1}{t}\right) \quad \text{(By Lemma 1)} \\
&= \sum_{t \in T(\mathbf{q})} \sum_{x \in B_t} p_x f\left(\frac{1}{t}\right) \\
&= \sum_{t \in T(\mathbf{q})} \sum_{x \in B_t} p_x f\left(\frac{1}{q_x}\right) \\
&= \ell(\mathbf{q}; \mathbf{p}).
\end{aligned}$$

If $f$ is strictly concave and there exists a $B_t$ where $\mathbf{q}$ and $\mathbf{p}$ disagree, then the inequality is strict.

## B  Additional Proofs for Strongly Proper Losses

### B.1  Proof of Theorem 2

Let us start by an analogous result to Lemma 1. We defer the proof of this lemma to Appendix B.2.

**Lemma 2.** *Suppose $f(z)$ is $b(z)$-left-strongly concave. Let $B \subseteq \mathcal{X}$ be any set and let $t(B) := \frac{\mathbf{p}(B)}{|B|}$,[5] and suppose $\sum_{x \in B} |p_x - t(B)| \geq \epsilon$. Let $\mu = \frac{1}{t(B)}$. Then*

$$\mathbb{E}_{X \sim \mathbf{p}}\left[f\left(\frac{1}{p_X}\right) \mid X \in B\right] \leq f(\mu) + \frac{b(\mu)}{32} \frac{\epsilon^2}{\mathbf{p}(B)^2 t(B)^2}.$$

*Proof of Theorem 2.* Note that a calibrated distribution $\mathbf{q}$ can be thought of as a piecewise uniform distribution with pieces $\{B_t\}_{t \in T(\mathbf{q})}$ and $\mathbf{q}(B_t) = \mathbf{p}(B_t)$. Let $\epsilon_t = \sum_{x \in B_t} |p_x - q_x|$, with

$\sum_{t \in T(\mathbf{q})} \epsilon_t = \epsilon = \|\mathbf{p} - \mathbf{q}\|_1$. Let $\alpha = \frac{\|\mathbf{p}-\mathbf{q}\|_1}{4}$ and let $H = \{t \in T(\mathbf{q}) : t \geq \frac{\alpha}{N}\}$ refer to indices of pieces in which the two distributions place reasonably high probability. We have:

$$\ell(\mathbf{q}; \mathbf{p}) - \ell(\mathbf{p}; \mathbf{p}) = \sum_x p_x \left[ f\left(\frac{1}{q_x}\right) - f\left(\frac{1}{p_x}\right) \right] = \sum_{t \in T(\mathbf{q})} \mathbf{p}(B_t) \left[ f\left(\frac{1}{t}\right) - \underset{X|B_t}{\mathbb{E}} \left[ f\left(\frac{1}{p_X}\right) \right] \right]$$

where $\mathbb{E}_{X|B_t}[\cdot]$ refers to the expectation over $X \sim \mathbf{p}$ conditioned on $X \in B_t$. Now consider any fixed component $B_t$. The difference inside the brackets is $f\left(\frac{1}{t}\right) - \mathbb{E}_{X|B_t}\left[ f\left(\frac{1}{p_X}\right)\right]$. Intuitively, strong concavity implies there should be a significant "Jensen gap". This is formalized in Lemma 2 of Appendix B that shows that if $\sum_{x \in B_j} |p_x - q_x| = \epsilon_j$, then

$$f\left(\frac{1}{t}\right) - \underset{X|B_t}{\mathbb{E}} \left[ f\left(\frac{1}{p_X}\right) \right] \geq \frac{b\left(\frac{1}{t}\right)}{32} \cdot \frac{\epsilon_t^2}{t^2 \mathbf{p}(B_t)^2}. \tag{2}$$

Summing over all $t \in T(\mathbf{q})$ and Applying the assumption that $b(x) \geq \frac{C(x)}{x^2}$ where $C(x)$ is nonincreasing along with the fact that $t \geq \frac{\alpha}{N}$ for $t \in H$ gives

$$\ell(\mathbf{q}; \mathbf{p}) - \ell(\mathbf{p}; \mathbf{p}) \geq \sum_{t \in T(\mathbf{q})} \mathbf{p}(B_t) \frac{b(\frac{1}{t})}{32} \frac{\epsilon_t^2}{t^2 \mathbf{p}(B_t)^2} \geq \sum_{t \in H} \mathbf{p}(B_t) \frac{b(\frac{1}{t})}{32} \frac{\epsilon_t^2}{t^2 \mathbf{p}(B_t)^2} \geq \frac{C\left(\frac{N}{\alpha}\right)}{32} \sum_{t \in H} \frac{\epsilon_t^2}{\mathbf{p}(B_t)}. \tag{3}$$

For $t \notin H$, since $\mathbf{q}(B_t) = \mathbf{p}(B_t) \leq \frac{\alpha|B_t|}{N}$ we have $\epsilon_t \leq \frac{2\alpha|B_t|}{N}$. Thus we have $\sum_{t \notin H} \epsilon_t \leq \frac{2\alpha}{N} |T(\mathbf{q}) \setminus H| \leq 2\alpha$, and so correspondingly, $\sum_{t \in H} \epsilon_t \geq \epsilon - 2\alpha$. Since the bound of (3) is increasing in each $\epsilon_t$ and decreasing in each $\mathbf{p}(B_t)$ we can obtain a lower bound by considering its minimum when $\sum_{t \in H} \epsilon_t = \epsilon - 2\alpha$ and $\sum_{t \in H} \mathbf{p}(B_t) = 1$. By the convexity of $(\cdot)^2$ this minimum is obtained at $\epsilon_t = \mathbf{p}(B_t) \cdot (\epsilon - 2\alpha)$.

This gives an overall bound of $\ell(\mathbf{q}; \mathbf{p}) - \ell(\mathbf{p}; \mathbf{p}) \geq \frac{C\left(\frac{N}{\alpha}\right)}{32} \cdot (\epsilon - 2\alpha)^2$. Replacing $\alpha = \frac{\|\mathbf{p}-\mathbf{q}\|_1}{4}$ in this bound completes theorem. $\square$

## B.2 Proof of Lemma 2

*Proof.* We draw $X \sim \mathbf{p}$ conditioned on $X \in B$. Let $S = \{x \in B : p_x > t(B)\}$. We upper-bound $f(\frac{1}{p_X})$ for each realization of $X$. If $p_X \leq t(B)$, then we simply use concavity. Otherwise, if $X \in S$, we use $b(z)$-left-strong-concavity. Furthermore, note that by Lemma 1, $\mathbb{E}_{X \sim \mathbf{p}|B}\left[\frac{1}{p_X}\right] = \mu$. We have:

$$\underset{X \sim \mathbf{p}|B}{\mathbb{E}} \left[ f\left(\frac{1}{p_X}\right) \right] \leq \mathbb{E}\left[ f(\mu) + df(\mu) \cdot \left(\frac{1}{p_X} - \mu\right) - \mathbf{1}\left[ X \in S \right] \frac{b(\mu)}{2} \left(\frac{1}{p_X} - \mu\right)^2 \right]$$

$$= f(\mu) - \frac{b(\mu)}{2} \frac{1}{\mathbf{p}(B)} \sum_{x \in S} p_x \left(\frac{1}{p_x} - \mu\right)^2,$$

Note the $\frac{1}{\mathbf{p}(B)}$ term arises from conditioning on $X \in B$. We now lower-bound the sum, using the constraint that $\sum_{x \in B} |p_x - t(B)| \geq \epsilon$, which implies that $\sum_{x \in S} p_x - t(B) \geq \frac{\epsilon}{2}$.

$$\sum_{x \in S} p_x \left(\frac{1}{t(B)} - \frac{1}{p_x}\right)^2 = \frac{\mathbf{p}(S)}{t(B)^2} - \frac{2|S|}{t(B)} + \sum_{x \in S} \frac{1}{p_x}.$$

Fixing $\mathbf{p}(S)$ and $|S|$, we get by convexity that this is minimized by $p_x$ constant on $S$, therefore equal to $t(B) + \frac{\epsilon}{2|S|}$. So we have

$$|S| \left( t(B) + \frac{\epsilon}{2|S|} \right) \left( \frac{1}{t(B)} - \frac{1}{t(B) + \frac{\epsilon}{2|S|}} \right)^2 = \left( |S| t(B) + \frac{\epsilon}{2} \right) \left( \frac{\epsilon}{2|S| \left( t(B)^2 + \frac{\epsilon t(B)}{2|S|} \right)} \right)^2$$

$$= \frac{|S| t(B) \epsilon^2 + \frac{\epsilon^3}{2}}{4|S|^2 t(B)^2 \left( t(B) + \frac{\epsilon}{2|S|} \right)^2}.$$

438 We consider the two cases for the larger term in the denominator. In the case $\frac{\epsilon}{2|S|} > t(B)$, we get

$$\geq \frac{|S|t(B)\epsilon^2 + \frac{\epsilon^3}{2}}{4|S|^2 t(B)^2 \left(\frac{\epsilon}{|S|}\right)^2}$$

$$\geq \frac{|S|t(B) + \frac{\epsilon}{2}}{4t(B)^2}$$

$$\geq \frac{\epsilon}{4t(B)^2}$$

$$\geq \frac{\epsilon^2}{4\mathbf{p}(B)t(B)^2}$$

439 where the last line follows because we must have $\epsilon \leq \mathbf{p}(B)$ from the definition of $\epsilon$. In the remaining
440 case, we get

$$\geq \frac{|S|t(B)\epsilon^2 + \frac{\epsilon^3}{2}}{4|S|^2 t(B)^2 \left(2t(B)\right)^2}$$

$$\geq \frac{\epsilon^2}{16|S|t(B)^3}$$

$$\geq \frac{\epsilon^2}{16|B|t(B)^3}$$

$$= \frac{\epsilon^2}{16\mathbf{p}(B)t(B)^2}.$$

441 $\qquad\qquad\qquad\qquad\qquad\qquad\qquad\qquad\qquad\qquad\qquad\qquad\qquad\qquad\qquad\qquad\square$

# C  Additional Proofs for Concentration of Losses

443 We first give a simple lemma that will be used to prove our main calibrated distribution concentration
444 result, Theorem 3.

445 **Lemma 3** (Calibrated Distribution Probability Lower Bound). *For any* $\mathbf{p} \in \Delta_{\mathcal{X}}$ *and* $\mathbf{q} \in \mathcal{C}(\mathbf{p})$, *for*
446 *any* $x \in \mathcal{X}$,

$$q_x \geq \frac{p_x}{N}. \qquad\qquad\qquad\qquad (4)$$

447 *Proof.* Let $B = \{x' : q_{x'} = q_x\}$. Then by calibration we have:

$$q_x = \frac{\mathbf{q}(B)}{|B|} \geq \frac{\mathbf{q}(B)}{N} = \frac{\mathbf{p}(B)}{N} \geq \frac{p_x}{N}.$$

448 $\qquad\qquad\qquad\qquad\qquad\qquad\qquad\qquad\qquad\qquad\qquad\qquad\qquad\qquad\qquad\qquad\square$

449 This bound is achieved when $\mathbf{q}$ is the uniform distribution and $\mathbf{p}$ is a point distribution.

## C.1  Proof of Theorem 3

451 We prove the following stronger result, Proposition 1, that only uses the lower-bound property
452 $q_x \geq \Omega(\frac{p_x}{N})$ and does not require a distribution to be calibrated. Combining this proposition with
453 Lemma 3 immediately proves Theorem 3.

454 **Proposition 1.** *Suppose* $\ell$ *is a local loss function with* $\ell(\mathbf{q}, x) = f\left(\frac{1}{q_x}\right)$ *for nonnegative, increasing,*
455 *concave* $f(z)$. *Suppose further that* $f(z) \leq cz^r$ *for all* $z \geq 1$, *some constant* $c > 0$, *and some constant*
456 $r < 1$. *Given* $\mathbf{p}$, *suppose* $\mathbf{q}$ *is* ***any*** *distribution such that* $q_x \geq \frac{c_2 p_x}{N}$ *for all* $x$ *and some constant*
457 $c_2 \in (0, 1]$. *Then, drawing at least* $m(\gamma, \delta, N)$ *samples guarantees that* $\left|\ell(\mathbf{q}; \hat{p}) - \ell(\mathbf{q}; \mathbf{p})\right| \leq \gamma$ *with*
458 *probability* $\geq 1 - \delta$ *if*

$$m(\gamma, \delta, N) \geq \frac{c_1 \cdot f(\beta)^2 \ln \frac{1}{\delta}}{\gamma^2},$$

459 *where* $c_1$ *is a fixed constant and* $\beta := \frac{2^{2/(1-r)} N^{3/(1-r)+2}}{c_2^{r/(1-r)} \delta \cdot \min(1, [\gamma/c]^{1/(1-r)})}$.

*Proof.* Fix a sample size $m \leq N$. Let $\Omega \subseteq \mathcal{X}$ be the set of $x$'s that occur with non-negligible probability:

$$\Omega = \left\{ x : p_x \geq \frac{c_2^{r/(1-r)} \cdot \delta \cdot \min(1, [\gamma/c]^{1/(1-r)})}{2^{2/(1-r)} N^{3/(1-r)+1}} \right\}.$$

we have $\mathbf{p}(\mathcal{X} \setminus \Omega) \leq N \cdot \frac{c_2^{r/(1-r)}\delta}{4N^4} \leq \frac{\delta}{4N}$ and thus for $x_1, \ldots x_m$ drawn i.i.d. from $\mathbf{p}$. By a union bound, letting $\mathcal{E}$ be the event that $x_1, \ldots, x_m \in \Omega$ and using that $m \leq N$:

$$\Pr[\mathcal{E}] \geq 1 - \frac{\delta}{4}. \tag{5}$$

We will condition on $\mathcal{E}$ going forward. First note that for $x \in \Omega$, we can bound $\ell(\mathbf{q}, x)$ using Lemma 3. Specifically, since $q_x \geq \frac{c_2 p_x}{N}$ and $f$ is nondecreasing, we have:

$$\ell(\mathbf{q}, x) = f\left(\frac{1}{q_x}\right) \leq f\left(\frac{2^{2/(1-r)} N^{3/(1-r)+2}}{c_2^{r/(1-r)} \cdot \delta \cdot \min(1, [\gamma/c]^{1/(1-r)})}\right).$$

Denote

$$\beta := \frac{2^{2/(1-r)} N^{3/(1-r)+2}}{c_2^{r/(1-r)} \cdot \delta \cdot \min(1, [\gamma/c]^{1/(1-r)})}.$$

Letting $z_i$ be the random variable

$$z_i = \frac{1}{m}\left(\ell(\mathbf{q}, x_i) - \mathop{\mathbb{E}}_{x \sim \mathbf{p}}[\ell(\mathbf{q}, x) | x \in \Omega]\right),$$

we have for $x_i \in \Omega$, $|z_i| \leq \frac{f(\beta)}{m}$ (where we use that $\ell(\mathbf{q}, x)$ is nonnegative by assumption.) So $\mathbb{E}[z_i^2 \mid x_i \in \Omega] \leq f(\beta)^2/m^2$. Then by a standard Bernstein inequality:

$$\Pr\left[\left|\frac{1}{m}\sum_{j=1}^{m}\ell(\mathbf{q}, x_j) - \mathop{\mathbb{E}}_{x \sim \mathbf{p}}[\ell(\mathbf{q}, x) | x \in \Omega]\right| \geq \frac{\gamma}{2} \mid \mathcal{E}\right] \leq \exp\left(-\frac{\gamma^2/8}{f(\beta)^2/m + f(\beta)/m \cdot \gamma/3}\right) \leq \frac{\delta}{2} \tag{6}$$

where the second inequality follows if we have $m \geq \frac{c_1 f(\beta)^2 \log(1/\delta)}{\gamma^2}$ for sufficiently large $c_1$. By a union bound, from (5) and (6) we have:

$$\Pr\left[\left|\frac{1}{m}\sum_{j=1}^{m}\ell(\mathbf{q}, x_j) - \mathop{\mathbb{E}}_{x \sim \mathbf{p}}[\ell(\mathbf{q}, x) | x \in \Omega]\right| \geq \frac{\gamma}{2}\right] \leq \delta.$$

It remains to show that the conditional expectation $\mathbb{E}_{x \sim \mathbf{p}}[\ell(\mathbf{q}, x) | x \in \Omega]$ is very close to $\ell(\mathbf{q}; \mathbf{p}) = \mathbb{E}_{x \sim \mathbf{p}}[\ell(\mathbf{q}, x)]$, which will give us the lemma. Intuitively, by conditioning on $x \in \Omega$ we are only removing very low probability events, which do not have a big effect on the loss. Specifically, we need to show that:

$$\left|\mathop{\mathbb{E}}_{x \sim \mathbf{p}}[\ell(\mathbf{q}, x) | x \in \Omega] - \ell(\mathbf{q}; \mathbf{p})\right| \leq \frac{\gamma}{2} \tag{7}$$

473  Since $\mathbf{p}(\mathcal{X} \setminus \Omega) \leq N \cdot \frac{c_2^{r/(1-r)}\delta \cdot \min(1,\gamma/c)}{4N^4} \leq \frac{c_2^{r/(1-r)}\gamma}{4N^3} \leq \frac{c_2^r\gamma}{4N^3}$, using that $f$ is nondecreasing,
474  $f(z) \leq cz^r$ for some $c$ and $r < 1$, and $q_x \geq \frac{c_2 p_x}{N}$:

$$
\begin{aligned}
\mathop{\mathbb{E}}_{x \sim \mathbf{p}}[\ell(\mathbf{q}, x) \mid x \in \Omega] &= \sum_{x \in \Omega} \frac{p_x}{\mathbf{p}(\Omega)} \cdot \ell(\mathbf{q}, x) \\
&\leq \frac{1}{1 - \frac{c_2^r \min(1,\gamma/c)}{4N^3}} \cdot \sum_{x \in \Omega} p_x \cdot \ell(\mathbf{q}, x) \\
&\leq \left(1 + \frac{c_2^r \min(1,\gamma/c)}{2N^3}\right) \cdot \sum_{x \in \mathcal{X}} p_x \cdot \ell(\mathbf{q}, x) \\
&\leq \ell(\mathbf{q};\mathbf{p}) + \frac{c_2^r \min(1,\gamma/c)}{2N^3} \cdot \sum_{x \in \mathcal{X}} p_x \cdot f\left(\frac{N}{c_2 p_x}\right) \\
&\leq \ell(\mathbf{q};\mathbf{p}) + \frac{c_2^r \min(1,\gamma/c)}{2N^3} \cdot c \cdot \frac{N^r}{c_2^r} \sum_{x \in \mathcal{X}} p_x^{1-r} \\
&= \ell(\mathbf{q};\mathbf{p}) + \frac{\min(1,\gamma/c)}{2N^3} \cdot c \cdot N^{2r} \\
&\leq \ell(\mathbf{q};\mathbf{p}) + \frac{\gamma}{2}.
\end{aligned}
$$

475  This gives us one side of (7). On the other side we have:

$$
\begin{aligned}
\mathop{\mathbb{E}}_{x \sim \mathbf{p}}[\ell(\mathbf{q}, x) \mid x \in \Omega] &= \sum_{x \in \Omega} \frac{p_x}{\mathbf{p}(\Omega)} \cdot \ell(\mathbf{q}, x) \\
&\geq \sum_{x \in \Omega} p_x \cdot \ell(\mathbf{q}, x) \\
&= \ell(\mathbf{q};\mathbf{p}) - \sum_{x \notin \Omega} p_x \cdot \ell(\mathbf{q}, x). \tag{8}
\end{aligned}
$$

476  Again using that $f(z) \leq cz^r$ for $r < 1$, that $q_x \geq \frac{c_2 p_x}{N}$, and that for $x \notin \Omega$ we have $p_x \leq$
477  $\frac{c_2^{r/(1-r)}\delta \cdot \min(1,[\gamma/c]^{1/(1-r)})}{2^{2/(1-r)}N^{3/(1-r)+1}}$:

$$
\sum_{x \notin \Omega} p_x \cdot \ell(\mathbf{q}, x) = \sum_{x \notin \Omega} p_x \cdot f\left(\frac{1}{q_x}\right) \leq c \sum_{x \notin \Omega} p_x^{1-r} \cdot \frac{N^r}{c_2^r} \leq c \cdot N^{r+1} \cdot \frac{\gamma/c}{4N^3} \leq \frac{\gamma}{4N}.
$$

478  Combined with (8) this yields the other side of (7), completing the bound and the proof.  □

479  *Proof of Theorem 3.*  Using Lemma 3, we have that for a calibrated distribution $\mathbf{q}$, $q_x \in \Omega(p_x/N)$
480  for all $x \in \mathcal{X}$. Together with the assumption that $f(z) \leq c\sqrt{z}$, we can directly apply Proposition 1 to
481  prove the claim.  □

# D  Additional Proofs for Sample Proper Losses

## D.1  Proof of Theorem 4

484  In the statement of Theorem 4 we require that $\ell(\mathbf{q}, x) = f\left(\frac{1}{q_x}\right)$ for $f$ that is nonnegative, increasing,
485  and $\frac{C(x)}{x^2}$-left-strongly concave. Further we require that $C(x)$ is non-decreasing and non-negative for
486  $x \geq 1$. Directly applying Theorem 2 we thus have:

$$
\ell(\mathbf{q};\mathbf{p}) - \ell(\mathbf{p};\mathbf{p}) \geq C\left(\frac{4N}{\|\mathbf{p}-\mathbf{q}\|_1}\right) \cdot \frac{\|\mathbf{p}-\mathbf{q}\|_1^2}{128}. \tag{9}
$$

487  Let $\gamma := C\left(\frac{4N}{\|\mathbf{p}-\mathbf{q}\|_1}\right) \cdot \frac{\|\mathbf{p}-\mathbf{q}\|_1^2}{128}$. Additionally, since $f(x) \leq c\sqrt{z}$ for $z \geq 1$ and since $\mathbf{q}, \mathbf{p} \in \mathcal{C}(\mathbf{p})$, ap-
488  plying Theorem 3 with error parameter $\gamma/3$ and failure parameter $\delta/2$, we have for $\beta := \frac{288N^8}{\delta \cdot \min(1,\gamma^2/c^2)}$,

if $m \geq \frac{c_1 f(\beta)^2 \lg \frac{2}{\delta}}{(\gamma/3)^2}$ for large enough constant $c_1$ then the following hold, each with probability $\geq 1 - \delta/2$:

$$|\ell(\mathbf{q}; \hat{\mathbf{p}}) - \ell(\mathbf{q}; \mathbf{p})| \leq \frac{\gamma}{3} \text{ and } |\ell(\mathbf{p}; \hat{\mathbf{p}}) - \ell(; \mathbf{p})| \leq \frac{\gamma}{3}.$$

By a union bound, with probability $\geq 1 - \delta$ both bounds hold simultaneously and by (9) we have:

$$\ell(\mathbf{q}; \hat{\mathbf{p}}) - \ell(\mathbf{p}; \hat{\mathbf{p}}) \geq \ell(\mathbf{q}; \mathbf{p}) - \ell(\mathbf{p}; \mathbf{p}) - \frac{2\gamma}{3} \geq \gamma - \frac{2\gamma}{3} > 0,$$

which completes the theorem. Plugging the value of $\gamma$ in we see that the bound holds for

$$m \geq \frac{c_1 f(\beta)^2 \ln \frac{1}{\delta}}{\left( C\left(\frac{4N}{\|\mathbf{p}-\mathbf{q}\|_1}\right) \cdot \frac{\|\mathbf{p}-\mathbf{q}\|_1^2}{128}/3 \right)^2} = \frac{c_1' f(\beta)^2 \ln \frac{1}{\delta}}{\left( C\left(\frac{4N}{\|\mathbf{p}-\mathbf{q}\|_1}\right) \cdot \|\mathbf{p}-\mathbf{q}\|_1^2 \right)^2}$$

for large enough constant $c_1'$. Additionally, we see that:

$$\beta = \frac{288 N^8}{\delta \cdot \min\left(1, \left[ C\left(\frac{4N}{\|\mathbf{p}-\mathbf{q}\|_1}\right) \cdot \frac{\|\mathbf{p}-\mathbf{q}\|_1^2}{128c} \right]^2 \right)}.$$

# E   Instantiation of Theorems 2, 3, and 4

Let us start with two observations regarding loss functions, characterizing inverse concave loss functions and inverse left-concave functions.

**Observation 1.** *Let $\ell(\mathbf{q}, x) = f\left(\frac{1}{q_x}\right)$ be such that $\ell$ is nonnegative, twice differentiable, decreasing, and convex. Then, $f(x)$ is concave.*

*Proof.* For ease of exposition, let $\ell(z) = f(\frac{1}{z})$.

$$\frac{df}{dy} = \frac{d\ell(\frac{1}{y})}{dz}\left(\frac{-1}{y^2}\right)$$

$$\frac{d^2 f}{dz^2} = \frac{d^2 \ell(\frac{1}{y})}{dz^2}\left(\frac{-1}{y^2}\right) + \frac{d\ell(\frac{1}{y})}{dz}\left(\frac{2}{y^3}\right).$$

Decreasing and convex gives a negative derivative and positive second derivative. Given that $y > 0$, we obtain a negative second derivative, hence concavity. $\square$

**Observation 2.** *Consider a nonincreasing function $b(z)$. A function $f$ is $b(z)$-left-strongly concave if for all $z$, $f''(z) \leq -b(z)$.*

*Proof.* We need to show that $f$ restricted to $[0, z]$ is $b(z)$-strongly concave. Consider $z_1 \geq z_2$. Since $b(z)$ is non-increasing we have for $t \in [z_2, z_1]$:

$$f'(t) = f'(z_2) + \int_{z_2}^{t} f''(s)ds \leq f'(z_2) - b(z) \cdot (t - z_2).$$

We thus have:

$$f(z_1) - f(z_2) = \int_{z_2}^{z_1} f'(t)dt \leq \int_{z_2}^{z_1} [f'(z_2) - b(z)(t - z_2)]dt$$

$$\leq f'(z_2) \cdot (z_1 - z_2) - b(z) \cdot \frac{(z_1 - z_2)^2}{2}.$$

Rearranging gives:

$$D_{-f}(z_1, z_2) := f(z_2) + f'(z_2) \cdot (z_1 - z_2) - f(z_1) \geq \frac{b(z)}{2} \cdot (z_1 - z_2)^2.$$

For $z_1 \leq z_2$, analogously for $t \in [z_1, z_2]$ we have:

$$f'(t) = f'(z_2) - \int_t^{z_2} f''(s)ds \geq f'(z_2) - b(z) \cdot (t - z_2)$$

and so

$$f(z_1) - f(z_2) = -\int_{z_2}^{z_1} f'(t)dt \leq \int_{z_2}^{z_1} [f'(z_2) - b(z)(t - z_2)]dt$$

$$\leq f'(z_2) \cdot (z_1 - z_2) - b(z) \cdot \frac{(z_1 - z_2)^2}{2}.$$

Rearranging gives again gives:

$$f(z_2) + f'(z_2) \cdot (z_1 - z_2) - f(z_1) \geq \frac{b(z)}{2} \cdot (z_1 - z_2)^2,$$

completing the lemma. $\qquad\square$

## E.1  Deriving Table 1

For $\ell(\mathbf{q}, x) = (\ln(1/q_x))^p$ for a constant $p \in (0, 1]$. By Observation 2, we have that $(\ln(z))^p$ is $C(z)/z^2$-left-strongly concave for

$$C(z) = p\ln(z)^{p-1} + p(1-p)\ln(z)^{p-2} \in \Theta\left(\ln(z)^{p-1}\right).$$

Moreover, $C(z)$ is non-increasing and non-negative for $z \geq 1$ and $\ln(z)^{p-1} \leq \sqrt{z}$. Using these, for any $\mathbf{p}$ and $\mathbf{q} \in \mathcal{C}(\mathbf{p})$ such that $\|\mathbf{p} - \mathbf{q}\|_1 \geq \epsilon$ we have

- By Theorem 2, $\ell(\mathbf{q}; \mathbf{p}) - \ell(\mathbf{p}; \mathbf{p}) = \Omega\left(\epsilon^2 \ln(N/\epsilon)^{p-1}\right)$.
- By Theorem 3, an empirical distribution $\hat{\mathbf{p}}$ of $O\left(\gamma^{-2}\ln(1/\delta)\ln(N/\delta\gamma)^{2p}\right)$ i.i.d samples from $\mathbf{p}$ is sufficient such that $|\ell(\mathbf{q}; \hat{\mathbf{p}}) - \ell(\mathbf{q}; \mathbf{p})| \leq \gamma$ with probability $1 - \delta$.
- By Theorem 4, an empirical distribution $\hat{\mathbf{p}}$ of

$$O\left(\frac{1}{\epsilon^4}\ln\left(\frac{1}{\delta}\right)\ln\left(\frac{N}{\delta\epsilon^2 \ln(N/\epsilon)^p}\right)^{2p}\ln(N/\epsilon)^{-2p+2}\right) \in O\left(\frac{1}{\epsilon^4}\ln\left(\frac{1}{\delta}\right)\ln\left(\frac{N}{\delta\epsilon}\right)^2\right)$$

  i.i.d samples from $\mathbf{p}$ is sufficient such that $\ell(\mathbf{q}; \hat{\mathbf{p}}) > \ell(\mathbf{p}; \hat{\mathbf{p}})$ with probability $1 - \delta$.

For $\ell(\mathbf{q}, x) = \ln(e^2/q_x)^2$. By Observation 2, we have that $\ln(e^2 \cdot z)^2$ is $\frac{2+2\ln(z)}{z^2}$-left-strongly concave. Since Theorem 2 requires that $C(z)$ is nonincreasing we cannot set $C(z) = 2 + 2\ln(z)$ as might be expected. Instead we set $C(z) = 2$. Additionally, using that $\ln(e^2 \cdot z)^2 \leq \sqrt{z}$, for any $\mathbf{p}$ and $\mathbf{q} \in \mathcal{C}(\mathbf{p})$ such that $\|\mathbf{p} - \mathbf{q}\|_1 \geq \epsilon$ we have

- By Theorem 2, $\ell(\mathbf{q}; \mathbf{p}) - \ell(\mathbf{p}; \mathbf{p}) = \Omega\left(\epsilon^2\right)$.
- By Theorem 3, an empirical distribution $\hat{\mathbf{p}}$ of $O\left(\gamma^{-2}\ln(1/\delta)\ln(N/\delta\gamma)^4\right)$ i.i.d samples from $\mathbf{p}$ is sufficient such that $|\ell(\mathbf{q}; \hat{\mathbf{p}}) - \ell(\mathbf{q}; \mathbf{p})| \leq \gamma$ with probability $1 - \delta$.
- By Theorem 4, an empirical distribution $\hat{\mathbf{p}}$ of

$$O\left(\frac{1}{\epsilon^4}\ln\left(\frac{1}{\delta}\right)\ln\left(\frac{N}{\delta\epsilon^2 \ln(N/\epsilon)}\right)^4\right) \in O\left(\frac{1}{\epsilon^4}\ln\left(\frac{1}{\delta}\right)\ln\left(\frac{N}{\delta\epsilon}\right)^4\right)$$

  i.i.d samples from $\mathbf{p}$ is sufficient such that $\ell(\mathbf{q}; \hat{\mathbf{p}}) > \ell(\mathbf{p}; \hat{\mathbf{p}})$ with probability $1 - \delta$.

## E.2  Other Loss Functions

We also instantiate Theorem 2 for a few natural loss functions that do not obtain strong finite sample bounds (Theorems 3, and 4).

For the linear loss $\ell_{lin\text{-}loss}(\mathbf{q}, x) = -q_x$, we have by Observation 2 that $-\frac{1}{z}$ is $\frac{2}{z^3}$-left-strongly-concave. Thus setting $C(z) = 1/z$, by Theorem 2 for any $\mathbf{p}$ and $\mathbf{q} \in \mathcal{C}(\mathbf{p})$ with $\|\mathbf{p} - \mathbf{q}\|_1 \geq \epsilon$:

$$\ell_{lin\text{-}loss}(\mathbf{q}; \mathbf{p}) - \ell_{lin\text{-}loss}(\mathbf{p}; \mathbf{p}) = \Omega\left(\frac{\epsilon}{N} \cdot \epsilon^2\right) = \Omega\left(\frac{\epsilon^3}{N}\right).$$

We can improve the dependence on $N$ and $\epsilon$ by considering e.g., $\ell(\mathbf{q}, x) = -\sqrt{q_x}$. In this case we have that $-1/\sqrt{z}$ is $\frac{3}{4z^{5/2}}$-left-strongly-concave. Thus setting $C(z) = \frac{3}{4\sqrt{z}}$, by Theorem 2 we have:

$$\ell(\mathbf{q}; \mathbf{p}) - \ell(\mathbf{p}; \mathbf{p}) = \Omega\left(\sqrt{\frac{\epsilon}{N}} \cdot \epsilon^2\right) = \Omega\left(\frac{\epsilon^{2.5}}{\sqrt{N}}\right).$$

 # F   Approximate Calibration

 In this section we show that our results are robust to a notion of approximate calibration and that we
 can construct distributions that satisfy approximate calibration using a small number of samples.

 **Definition 8** (Approximate Calibration). For $\mathbf{q} \in \Delta_{\mathcal{X}}$, for any $t \in [0,1]$, let $B_t = \{x : q_t = t\}$. $\mathbf{q}$
 is $(\alpha_1, \alpha_2)$-*approximately calibrated with respect to* $\mathbf{p}$ if there is some subset $T \subseteq [0,1]$ such that
 $\mathbf{q}(B_t) \in (1 \pm \alpha_1)\mathbf{p}(B_t)$ for all $t \notin T$, $\mathbf{q}(B_t) \geq (1-\alpha_1)\mathbf{p}(B_t)$ for all $t \in T$, and $\mathbf{q}(\cup_{t \in T} B_t) \leq \alpha_2$.
 Let $\mathcal{C}(\mathbf{p}, \alpha_1, \alpha_2)$ denote the set of all $(\alpha_1, \alpha_2)$-approximately calibrated distributions w.r.t. $\mathbf{p}$.

 Intuitively, $\mathbf{q} \in \mathcal{C}(\mathbf{p}, \alpha_1, \alpha_2)$ is calibrated up to $(1 \pm \alpha_1)$ multiplicative error on any bucket $B_t$ where
 $\mathbf{q}$ and hence $\mathbf{p}$ place reasonably large mass. There is some set of buckets (corresponding to $t \in T$)
 where $\mathbf{q}$ may significantly overestimate the probability assigned by $\mathbf{p}$, however, the total mass placed
 on these buckets will still be small – at most $\alpha_2$.

 ## F.1   Efficiently Constructing Approximately Calibrated Distributions

 We now demonstrate that, given a candidate distribution $\mathbf{q}$ and sample access to $\mathbf{p}$, it is possible
 to efficiently construct $\mathbf{q}' \in \mathcal{C}(\mathbf{p}, \alpha_1, \alpha_2)$. Further, if $\mathbf{q} \in \mathcal{C}(\mathbf{p}, \alpha_1, \alpha_2)$ we will have $\|\mathbf{q} - \mathbf{q}'\|_1 \leq$
 $O(\alpha_1 + \alpha_2)$. In this way, if $\mathbf{q}$ is approximately calibrated, we can certify at least that it is close
 to another approximately calibrated distribution. Of $\mathbf{q}$ is not approximately calibrated, we return a
 distribution that is approximately calibrated, which of course, may be far from $\mathbf{q}$.

 **Theorem 5.** *Given any* $\mathbf{q} \in \Delta_{\mathcal{X}}$, *sample access to* $\mathbf{p} \in \Delta_{\mathcal{X}}$, *and parameters* $\alpha_1, \alpha_2, \delta \in (0,1]$ *there*
 *is an algorithm that takes* $O\left( \frac{\log\left(\frac{N}{\alpha_1}\right)^2 \cdot \log\left(\frac{\log N}{\delta \alpha_1}\right)}{\alpha_1^4 \cdot \alpha_2^2} \right)$ *samples from* $\mathbf{p}$ *and returns, with probability*
 $\geq 1 - \delta$, $\mathbf{q}' \in \mathcal{C}(\mathbf{p}, \alpha_1, \alpha_2)$. *Further, if* $\mathbf{q} \in \mathcal{C}(\mathbf{p}, \alpha_1, \alpha_2)$ *then* $\|\mathbf{q} - \mathbf{q}'\|_1 \leq O(\alpha_1 + \alpha_2)$.

 The main idea of the algorithm achieving Theorem 5 is to round $\mathbf{q}$'s probabilities into buckets of
 multiplicative width $(1 \pm \alpha_1)$. We can then efficiently approximate the total probability mass in each
 bucket, excluding those that may have very small mass. On these buckets, we may over approximate
 the true mass, and thus they are included in the set $T$ in Definition 8.

 We start with a simple lemma that shows, using a standard concentration bound, how well we can
 approximate the probability of any event under any distribution.

 **Lemma 4.** *For any* $\mathbf{p} \in \Delta_{\mathcal{X}}$ *and* $B \subseteq \mathcal{X}$, *given* $m$ *independent samples* $x_1, \ldots x_m \sim \mathbf{p}$, *there is*
 *some fixed constant* $c$ *such that, for any* $\epsilon, \delta \in (0,1]$, *if* $m \geq \frac{3 \ln(2/\delta)}{\epsilon^2}$, *then with probability* $\geq 1 - \delta$:

$$\left| \mathbf{p}(B) - \frac{|\{x_i : x_i \in B\}|}{m} \right| \leq \epsilon.$$

 *Proof.* $\mathbb{E}\, |\{x_i : x_i \in B\}| = m \cdot \mathbf{p}(B)$. By a standard Chernoff bound:

$$\Pr\left[ |\,|\{x_i : x_i \in B\}| - m \cdot \mathbf{p}(B)| \geq m \cdot \epsilon \right] \leq e^{-\frac{\left(\frac{\epsilon}{\mathbf{p}(B)}\right)^2}{2 + \frac{\epsilon}{\mathbf{p}(B)}} m\mathbf{p}(B)} + e^{-\frac{\left(\frac{\epsilon}{\mathbf{p}(B)}\right)^2}{2} m\mathbf{p}(B_i)}$$
$$\leq e^{-\frac{\epsilon^2 m}{2\mathbf{p}(B) + \epsilon}} + e^{-\frac{\epsilon^2 m}{2}}$$
$$\leq 2e^{-\frac{\epsilon^2 m}{3}},$$

 which is $\leq \delta$ as long as $m \geq \frac{3\ln(2/\delta)}{\epsilon^2}$. $\qquad\square$

 With Lemma 4 in hand, we proceed to the proof of Theorem 5.

*Proof of Theorem 5.* For convenience, define $\gamma_1 = \frac{\alpha_1}{3}$, and $b = \lceil \log_{1-\frac{\gamma_1}{8}} \frac{\gamma_1}{8N} \rceil$. Note that $b =$
$O\left( \frac{\log \frac{N}{\alpha_1}}{\alpha_1} \right)$. For $i \in \{1, \ldots, b\}$, define:

$$\bar{B}_i = \left\{ x : q_x \in \left( \left(1 - \frac{\gamma_1}{8}\right)^i, \left(1 - \frac{\gamma_1}{8}\right)^{i-1} \right] \right\}.$$

Let $\bar{B}_{b+1} = \left\{ x : q_x \leq \left(1 - \frac{\gamma_1}{8}\right)^b \right\}$.[6] Note that $\bar{B}_1 \cup \ldots \cup \bar{B}_b \cup \bar{B}_{b+1} = \mathcal{X}$. Now, via Lemma 4, with $O\left(\frac{b^2 \cdot \log b/\delta}{\alpha_2^2 \cdot \alpha_1^2}\right)$ samples from $\mathbf{p}$ it is possible to compute $\tilde{\mathbf{p}}(\bar{B}_1), \ldots, \tilde{\mathbf{p}}(\bar{B}_{b+1})$ such that, with probability $\geq 1 - \delta$,

$$|\mathbf{p}(\bar{B}_i) - \tilde{\mathbf{p}}(\bar{B}_i)| \leq \frac{\gamma_1 \cdot \alpha_2}{8(b+1)}$$

565  for all $i$ simultaneously. Let $\mathcal{E}$ be the event that these approximations hold, and assume that $\mathcal{E}$ occurs.
566  Then for any $i$ with $\tilde{\mathbf{p}}(\bar{B}_i) \leq \frac{\alpha_2}{4(b+1)}$, it must be that

$$\mathbf{p}(\bar{B}_i) \leq \frac{\alpha_2}{4(b+1)} + \frac{\gamma_1 \cdot \alpha_2}{8(b+1)} \leq \frac{\alpha_2}{2(b+1)}. \tag{10}$$

567  Let $L \subseteq \{1, \ldots, b+1\}$ be the set of all such $i$. Similarly, for $i$ with $\tilde{\mathbf{p}}(\bar{B}_i) > \frac{\alpha_2}{4(b+1)}$, it must be that:

$$\mathbf{p}(B_i) > \frac{\alpha_2}{4(b+1)} - \frac{\gamma_1 \cdot \alpha_2}{8(b+1)} > \frac{\alpha_2}{8(b+1)}. \tag{11}$$

568  Let $H = \{1, \ldots, b+1\} \setminus L$ be the set of all such $i$.

569  Define $\mathbf{w}$ as follows: for $x \in \cup_{i \in L} \bar{B}_i$ set $w_x = \frac{\alpha_2}{2|\cup_{i \in L} \bar{B}_i|}$. For $i \in H$, for $x \in \bar{B}_i$ let $w_x = \frac{\tilde{\mathbf{p}}(\bar{B}_i)}{|\bar{B}_i|}$.
570  We have the following facts about $\mathbf{w}$:

571  1. For $i \in H$, $\mathbf{w}(\bar{B}_i) = \tilde{\mathbf{p}}(\bar{B}_i) \in \mathbf{p}(\bar{B}_i) \pm \frac{\gamma_1 \cdot \alpha_2}{8(b+1)}$, which by the fact that $\mathbf{p}(\bar{B}_i) \geq \frac{\alpha_2}{8(b+1)}$
572  (equation (11)) gives for all $i \in H$:

$$\mathbf{w}(\bar{B}_i) \in (1 \pm \gamma_1) \mathbf{p}(\bar{B}_i). \tag{12}$$

573  2. $\mathbf{w}(\cup_{i \in L} B_i) = \frac{\alpha_2}{2}$ and by (10), $\mathbf{p}(\cup_{i \in L} \bar{B}_i) = \sum_{i \in L} \mathbf{p}(\bar{B}_i) \leq (b+1) \cdot \frac{\alpha_2}{2(b+1)} = \frac{\alpha_2}{2}$.

574  In combination, the above facts give that $\|\mathbf{w}\|_1 \in (1 \pm \gamma_1)$. Thus, letting $\mathbf{q}' = \frac{1}{\|\mathbf{w}\|_1} \cdot \mathbf{w}$, we have:

575  1. Applying (12), for all $i \in H$, $\left(\frac{1-\gamma_1}{1+\gamma_1}\right) \mathbf{p}(\bar{B}_i) \leq \mathbf{q}'(\bar{B}_i) \leq \left(\frac{1+\gamma_1}{1-\gamma_1}\right) \mathbf{p}(\bar{B}_i)$. Since $\gamma_1 = \frac{\alpha_1}{3}$
576  we have $\frac{1-\gamma_1}{1+\gamma_1} \geq 1 - \alpha_1$ and $\frac{1+\gamma_1}{1-\gamma_1} \leq 1 + \alpha_1$, which gives for all $i \in H$:

$$\mathbf{q}'(\bar{B}_i) \in (1 \pm \alpha_1) \mathbf{p}(\bar{B}_i). \tag{13}$$

577  2. $\mathbf{q}'(\cup_{i \in L} \bar{B}_i) \geq \frac{1}{1+\gamma_1} \cdot \frac{\alpha_2}{2} \geq (1 - \alpha_1) \cdot \frac{\alpha_2}{2} \geq (1 - \alpha_1) \cdot \mathbf{p}(\cup_{i \in L} \bar{B}_i)$. Additionally,
578  $\mathbf{q}'(\cup_{i \in L} \bar{B}_i) \leq \frac{1}{1-\gamma_1} \cdot \frac{\alpha_2}{2} \leq \alpha_2$.
579  3. $\|\mathbf{q}' - \mathbf{w}\|_1 \leq \gamma_1$.

580  Properties (1) and (2) together give that $\mathbf{q}' \in \mathcal{C}(\mathbf{p}, \alpha_1, \alpha_2)$ where we define the set $T$ to be $\{\bar{q}_x\}$ for
581  $x \in \cup_{i \in L} \bar{B}_i$.

582  Recalling that $b = O\left(\frac{\log N/\alpha_1}{\alpha_1}\right)$, the overall sample complexity used to construct $\mathbf{q}'$ is:

$$O\left(\frac{b^2 \cdot \log b/\delta}{\alpha_2^2 \cdot \alpha_1^2}\right) = O\left(\frac{\log(N/\alpha_1)^2 \log(b/\delta)}{\alpha_1^4 \cdot \alpha_2^2}\right) = O\left(\frac{\log(N/\alpha_1)^2 \cdot \log\left(\frac{\log N}{\delta \alpha_1}\right)}{\alpha_1^4 \cdot \alpha_2^2}\right).$$

583  Finally, it remains to show that if $\mathbf{q} \in \mathcal{C}(\mathbf{p}, \alpha_1, \alpha_2)$, then $\|\mathbf{q} - \mathbf{q}'\|_1 \leq O(\alpha_1 + \alpha_2)$.

584  For every $j \leq b$, since $\mathbf{q}$ places all probabilities within $(1 \pm \frac{\gamma_1}{8}) = (1 \pm \frac{\alpha_1}{24})$ of each other on this
585  bucket, for every $x \in \bar{B}_j$, $q_x \in (1 \pm \frac{\alpha_1}{24}) \cdot \frac{\mathbf{q}(\bar{B}_j)}{|\bar{B}_j|}$. We thus have:

$$\sum_{x \in \bar{B}_j} |q_x - q'_x| \leq |\mathbf{q}(\bar{B}_j) - \mathbf{q}'(\bar{B}_j)| + O(\alpha_1) \cdot \mathbf{q}(\bar{B}_j).$$

For $\bar{B}_{b+1}$ since $\mathbf{q}(\bar{B}_{b+1}) \leq \frac{\alpha}{24}$, we simply have $\sum_{x \in \bar{B}_{b+1}} |q_x - q'_x| \leq |\mathbf{q}(\bar{B}_j) - \mathbf{q}'(\bar{B}_j)| + O(\alpha_1)$. Thus overall:

$$\|\mathbf{q} - \mathbf{q}'\|_1 = \sum_{j=1}^{b+1} \sum_{x \in \bar{B}_j} |q_x - q'_x| \leq \sum_{j=1}^{b+1} |\mathbf{q}(\bar{B}_j) - \mathbf{q}'(\bar{B}_j)| + O(\alpha_1).$$

We now bound the above sum using that both $\mathbf{q}$ and $\mathbf{q}'$ are in $\mathcal{C}(\mathbf{p}, \alpha_1, \alpha_2)$. Let $T$ be the set of probabilities for which $\mathbf{q}$ may significantly overestimate $\mathbf{p}$ but places mass $\leq \alpha_2$. Let $T'$ be analogous set for $\mathbf{q}'$ (see Definition 8). Let $\bar{\mathbf{q}}$ be vector obtained by setting $q_x = p_x$ for $\{x : q_x \in T\}$. Let $\bar{\mathbf{q}}'$ be defined analogously for $\mathbf{q}'$. We have:

$$\|\mathbf{q} - \mathbf{q}'\|_1 \leq \sum_{j=1}^{b+1} |\mathbf{q}(\bar{B}_j) - \mathbf{q}'(\bar{B}_j)| + O(\alpha_1) \leq \sum_{j=1}^{b+1} |\bar{\mathbf{q}}(\bar{B}_j) - \bar{\mathbf{q}}'(\bar{B}_j)| + O(\alpha_1 + \alpha_2).$$

Additionally, we can see that both $\bar{\mathbf{q}}$ and $\bar{\mathbf{q}}'$ are calibrated up to error $(1 \pm \alpha_1)$ on all $\bar{B}_j$ ($\bar{\mathbf{q}}$ is calibrated up to this error on all its level sets, which form a refinement of $\{\bar{B}_j\}$.) Thus we have:

$$\|\mathbf{q} - \mathbf{q}'\|_1 \leq \sum_{j=1}^{b+1} O(\alpha_1) \cdot \mathbf{p}(\bar{B}_j) + O(\alpha_1 + \alpha_2) = O(\alpha_1 + \alpha_2).$$

which completes the claim. $\qquad\square$

## F.2 Strong Properness Under Approximate Calibration

We now show that Theorem 2 is robust to approximation calibration, using a similar proof strategy. See Table 2 for a sampling of results that this implies, which essentially match those given by Table 1 in the case of exact calibration.

**Theorem 6.** *Suppose* $\ell(\mathbf{q}, x) = f(\frac{1}{q_x})$ *where* $f$ *is non-decreasing, and for* $z \geq \frac{1}{\max_x q_x}$ *is non-negative and satisfies* $f'(z) \leq \frac{D(z)}{z}$ *for some non-decreasing function* $D$. *Also suppose that* $f$ *is* $\frac{C(z)}{z^2}$-*left-strongly concave for* $C$ *that is non-increasing and non-negative for* $z \geq 1$. *Then for all* $p \in \Delta_{\mathcal{X}}$, $\alpha_1 \leq 1/2$ *and* $\mathbf{q} \in \mathcal{C}(\mathbf{p}, \alpha_1, \alpha_2)$:

$$\ell(\mathbf{q}; \mathbf{p}) - \ell(\mathbf{p}; \mathbf{p}) \geq \frac{C\left(\frac{N}{2\alpha_2}\right)}{32} \cdot (\|\mathbf{p} - \mathbf{q}\|_1 - \alpha_1 - 5\alpha_2)^2 - 2\alpha_1 \cdot D\left(\frac{N}{2\alpha_2}\right) - 3\alpha_2 \cdot f\left(\frac{N}{3\alpha_2}\right).$$

*Proof.* Let $\mathbf{q} \in \mathcal{C}(\mathbf{p}, \alpha_1, \alpha_2)$ be piecewise uniform with pieces $\{B_t\}_{t \in T(\mathbf{q})}$. Let $L_1 = \left\{ t : \frac{\mathbf{p}(B_t)}{|B_t|} \leq \frac{\alpha_2}{N} \right\}$. Let $H \subseteq T(\mathbf{q}) \setminus L_1$ contain all remaining $t$ for which $\mathbf{q}(B_t) \in (1 \pm \alpha_1)\mathbf{p}(B_t)$. Finally, let $L_2 = T(\mathbf{q}) \setminus (H \cup L_1)$ contain all remaining $t \in T(\mathbf{q})$. Let $\epsilon_t = \sum_{x \in B_t} |p_x - q_x|$, with $\sum_{t \in T(\mathbf{q})} \epsilon_t = \epsilon = \|\mathbf{p} - \mathbf{q}\|_1$. Finally, consider $\mathbf{q}' \in \mathcal{C}(\mathbf{p})$ that is exactly calibrated and piecewise uniform on $B_t(\mathbf{q})$, that is, $q'_x = \mathbf{p}(B_t(\mathbf{q}))/|B_t|$ for all $x \in B_t(\mathbf{q})$ and $t \in T(\mathbf{q})$.

By definition of $L_1$ we have $\mathbf{p}(\cup_{t \in L_1} B_t) = \mathbf{q}'(\cup_{t \in L_1} B_t) \leq \alpha_2$. Additionally, by our definition of approximate calibration, for any $t \in L_1$, either $\mathbf{q}(B_t) \in (1 \pm \alpha_1)\mathbf{p}(B_t)$ or else $t \in T$ is in the set of buckets for which the total mass $\mathbf{q}(\cup_{t \in T} B_t) \leq \alpha_2$. We have

$$\mathbf{q}(\cup_{t \in L_1} B_t) \leq (1 + \alpha_1)\alpha_2 + \alpha_2 \leq 3\alpha_2.$$

Similarly, using the definition of approximate calibration we have:

$$\mathbf{q}(\cup_{t \in L_2} B_t) \leq \alpha_2 \text{ and } \mathbf{p}(\cup_{t \in L_2} B_t) = \mathbf{q}'(\cup_{t \in L_2} B_t) \leq \frac{\alpha_2}{1 - \alpha_1} \leq 2\alpha_2.$$

This gives us that the truly calibrated $\mathbf{q}'$ is close to the approximately calibrated $\mathbf{q}$:

$$\|\mathbf{q} - \mathbf{q}'\|_1 \leq \sum_{t \in H} \alpha_1 \cdot \mathbf{p}(B_t) + \sum_{t \in L_1 \cup L_2} (\mathbf{q}(B_t) + \mathbf{q}'(B_t)) \leq \alpha_1 + 5\alpha_2.$$

Thus, by triangle inequality we have

$$\|\mathbf{p} - \mathbf{q}'\|_1 \geq \|\mathbf{p} - \mathbf{q}\|_1 - \alpha_1 - 5\alpha_2. \tag{14}$$

We can thus bound $\ell(\mathbf{q}'; \mathbf{p}) - \ell(\mathbf{p}; \mathbf{p})$ following the proof of Theorem 2. Let $\epsilon' = \|\mathbf{p} - \mathbf{q}'\|_1$ and $\epsilon'_t = \sum_{x \in B_t} |p_x - q'_x|$. Let $\ell_H(\mathbf{q}; \mathbf{p}) = \sum_{j \in H} \sum_{x \in B_t} p_x f\left(\frac{1}{q_x}\right)$ be the loss restricted to the buckets in $H$. By (2) we can bound:

$$\ell(\mathbf{q}'; \mathbf{p}) - \ell(\mathbf{p}; \mathbf{p}) \geq \ell_H(\mathbf{q}'; \mathbf{p}) - \ell_H(\mathbf{p}; \mathbf{p}) \geq \sum_{t \in H} \mathbf{p}(B_t) \frac{b\left(\frac{|B_t|}{\mathbf{q}'(B_t)}\right)}{32} \frac{(\epsilon'_t)^2}{\left(\frac{\mathbf{q}'(B_t)}{|B_t|}\right)^2 \mathbf{p}(B_t)^2}.$$

Since $H$ excludes call elements in $L_1$, for all $t \in H$, $\frac{\mathbf{q}'(B_t)}{|B_t|} \geq \frac{\alpha_2}{N}$. Thus by our assumption on $b(\cdot)$:

$$\ell_H(\mathbf{q}'; \mathbf{p}) - \ell_H(\mathbf{p}; \mathbf{p}) \geq \sum_{t \in H} \frac{C\left(\frac{N}{\alpha_2}\right)}{32} \frac{(\epsilon'_t)^2}{\mathbf{p}(B_t)}.$$

and applying the same argument as in Theorem 2 can lower bound this quantity using (14) by:

$$\ell_H(\mathbf{q}'; \mathbf{p}) - \ell_H(\mathbf{p}; \mathbf{p}) \geq \frac{C\left(\frac{N}{\alpha_2}\right)}{32} \cdot \left(\|\mathbf{p} - \mathbf{q}\|_1 - \alpha_1 - 5\alpha_2\right)^2. \tag{15}$$

We next show that $\ell_H(\mathbf{q}'; \mathbf{p}) - \ell_H(\mathbf{q}; \mathbf{p})$ is not too large. Since $\mathbf{q}$ and $\mathbf{q}'$ are both piecewise uniform on $\{B_t\}_{t \in T(\mathbf{q})}$ and since $\mathbf{q}'$ is calibrated (i.e, $\mathbf{q}'(B_t) = \mathbf{p}(B_t)$ for all $t$),

$$\ell_H(\mathbf{q}'; \mathbf{p}) - \ell_H(\mathbf{q}; \mathbf{p}) = \ell_H(\mathbf{q}'; \mathbf{q}') - \ell_H(\mathbf{q}; \mathbf{q}').$$

We have using that $f$ is nondecreasing:

$$\ell_H(\mathbf{q}'; \mathbf{q}') = \sum_{t \in H} \sum_{x \in B_t} \mathbf{q}'(B_t) \cdot f\left(\frac{|B_t|}{\mathbf{q}'(B_t)}\right) \leq \sum_{t \in H} \sum_{x \in B_t} \mathbf{q}'(B_t) \cdot f\left(\frac{1}{(1 - \alpha_1) \cdot q_x}\right) \tag{16}$$

Using the concavity of $f$ along with the assumption that $f'(z) \leq \frac{D(z)}{z}$, we have:

$$f\left(\frac{1}{(1 - \alpha_1) \cdot q_x}\right) \leq f\left(\frac{1}{q_x}\right) + f'\left(\frac{1}{q_x}\right) \cdot \left(\frac{1}{(1 - \alpha_1) q_x} - \frac{1}{q_x}\right)$$

$$\leq f\left(\frac{1}{q_x}\right) + D\left(\frac{1}{q_x}\right) \cdot q_x \cdot \frac{\alpha_1}{(1 - \alpha_1) q_x}$$

$$\leq f\left(\frac{1}{q_x}\right) + D\left(\frac{1}{q_x}\right) \cdot 2\alpha_1.$$

Plugging back into (16), using that $q_x \geq (1 - \alpha)q'_x \geq \frac{\alpha_2(1 - \alpha_1)}{N} \geq \frac{\alpha_2}{2N}$ for all $x \in \cup_{t \in H} B_t$ we have:

$$\ell_H(\mathbf{q}'; \mathbf{q}') \leq \sum_{t \in H} \sum_{x \in B_t} \mathbf{q}'(B_t) \left[ f\left(\frac{1}{q_x}\right) + D\left(\frac{N}{2\alpha_2}\right) \cdot 2\alpha_1 \right]$$

$$\leq \ell_H(\mathbf{q}; \mathbf{q}') + D\left(\frac{N}{2\alpha_2}\right) \cdot 2\alpha_1.$$

Combined with (15) this gives:

$$\ell_H(\mathbf{q}; \mathbf{p}) - \ell_H(\mathbf{p}; \mathbf{p}) \geq \frac{C\left(\frac{N}{2\alpha_2}\right)}{32} \cdot \left(\|\mathbf{p} - \mathbf{q}\|_1 - \alpha_1 - 5\alpha_2\right)^2 - 2\alpha_1 \cdot D\left(\frac{N}{2\alpha_2}\right). \tag{17}$$

Finally, let $\ell_L(\mathbf{q}; \mathbf{p})$ be the loss restricted to buckets in $L_1 \cup L_2$. As shown, $\sum_{t \in L_1 \cup L_2} \sum_{x \in B_t} p_x \leq 3\alpha_2$. By the concavity of $f(z)$ we thus have:

$$\ell_L(\mathbf{p}; \mathbf{p}) = \sum_{t \in L_1 \cup L_2} \sum_{x \in B_t} p_x \cdot f\left(\frac{1}{p_x}\right) \leq 3\alpha_2 \cdot f\left(\frac{N}{3\alpha_2}\right).$$

627 Combined with (17) this finally gives:

$$\ell(\mathbf{q};\mathbf{p}) - \ell(\mathbf{p};\mathbf{p}) \geq \ell_H(\mathbf{q};\mathbf{p}) - \ell_H(\mathbf{p};\mathbf{p}) - \ell_L(\mathbf{p};\mathbf{p})$$

$$\geq \frac{C\left(\frac{N}{2\alpha_2}\right)}{32} \cdot \left(\|\mathbf{p} - \mathbf{q}\|_1 - \alpha_1 - 5\alpha_2\right)^2 - 2\alpha_1 \cdot D\left(\frac{N}{2\alpha_2}\right) - 3\alpha_2 \cdot f\left(\frac{N}{3\alpha_2}\right),$$

628 which completes the theorem. □

| $\ell(\mathbf{q}, x)$ | $f(z)$ | $D(z)$ | $C(z)$ | $\alpha_1$ | $\alpha_2$ | $\frac{\ell(\mathbf{q};\mathbf{p})-\ell(\mathbf{p};\mathbf{p})}{\epsilon^2}$ |
|---|---|---|---|---|---|---|
| $\ln\frac{1}{q_x}$ | $\ln(z)$ | $1$ | $1$ | $\Theta\left(\epsilon^2\right)$ | $\Theta\left(\frac{\epsilon^2}{\ln N}\right)$ | $\Omega(1)$ |
| $\ln\frac{1}{q_x}^p, p \in (0,1]$ | $(\ln(z))^p$ | $1$ | $\ln(z)^{p-1}$ | $\Theta\left(\epsilon^2\right)$ | $\Theta\left(\frac{\epsilon^2}{(\ln N)^p}\right)$ | $\Omega\left((\ln N)^{p-1}\right)$ |
| $\ln\left(\ln\frac{1}{q_x}\right)$ | $\ln(\ln(z))$ | $1$ | $1/\ln(z)$ | $\Theta\left(\epsilon^2\right)$ | $\Theta\left(\frac{\epsilon^2}{\ln(\ln N)}\right)$ | $\Omega\left(\frac{1}{\ln N}\right)$ |
| $\frac{1}{\sqrt{q_x}}$ | $\sqrt{z}$ | $2\sqrt{z}$ | $\frac{1}{4\sqrt{z}}$ | $\Theta\left(\frac{\epsilon^4}{N}\right)$ | $\Theta\left(\frac{\epsilon^4}{N}\right)$ | $\Omega\left(\frac{\epsilon^2}{N}\right)$ |
| $\left(\ln\frac{e^2}{q_x}\right)^2$ | $\ln(e^2 z)^2$ | $2\ln(z)+2$ | $2$ | $\Theta\left(\frac{\epsilon^2}{\ln N}\right)$ | $\Theta\left(\frac{\epsilon^2}{(\ln N)^2}\right)$ | $\Omega(1)$ |

Table 2: Examples of loss functions that are strongly proper over $\mathcal{C}(\mathbf{p}, \alpha_1, \alpha_2)$. We let $\epsilon := \|\mathbf{p} - \mathbf{q}\|_1$ and assume for simplicity that $\epsilon \geq 1/N$. We fix values of $\alpha_1$ and $\alpha_2$ that yield a strong properness bound nearly matching that of Theorem 3 for truly calibrated distributions. Note that in the theorem $D(z)$ is required to be nondecreasing and thus we set it to 1 for all loss functions considered that grow slower than the log loss.

629 ## F.3 Concentration Under Approximate Calibration

630 It is also easy to show that our main concentration result, Theorem 3, is robust to approximate
631 calibration, since this result just uses that calibration ensures $\frac{q_x}{p_x}$ is not too small for any $x$ (Lemma
632 3). In particular, using an identical argument to what is used in Lemma 3 we can see from Definition
633 8 that for $\mathbf{q} \in \mathcal{C}(\mathbf{p}, \alpha_1, \alpha_2)$, for all $x$, $q_x \geq \frac{(1-\alpha_1)p_x}{N} \geq \frac{p_x}{2N}$ for $\alpha_1 \leq 1/2$. Following the proof of
634 Theorem 3 using this bound in place of Lemma 3 gives:

**Theorem 7.** *Suppose $\ell$ is a local loss function with $\ell(\mathbf{q}, x) = f\left(\frac{1}{q_x}\right)$ for non-negative, non-decreasing, concave $f(z)$. Suppose further that $f(z) \leq c\sqrt{z}$ for all $z \geq 1$ and some constant $c$. Then $\ell$ concentrates over $\mathcal{C}(\mathbf{p}, \alpha_1, \alpha_2)$ for any $\alpha_1 \leq 1/2$ and $m(\gamma, \delta, N) \leq N$ satisfying*

$$m(\gamma, \delta, N) \geq \frac{c_1 \cdot f(\beta)^2 \ln\frac{1}{\delta}}{\gamma^2},$$

635 *where $c_1$ is a fixed constant and $\beta := \frac{32N^8}{\delta \cdot \min(1, \gamma^2/c^2)}$.*

636 *That is, for any $\mathbf{p} \in \Delta_{\mathcal{X}}, \mathbf{q} \in \mathcal{C}(\mathbf{p}, \alpha_1, \alpha_2)$, drawing at least $m(\gamma, \delta, N)$ samples guarantees*
637 *$|\ell(\mathbf{q}; \hat{\mathbf{p}}) - \ell(\mathbf{q}; \mathbf{p})| \leq \gamma$ with probability $\geq 1 - \delta$.*

638 First, the analogue of Lemma 3.

639 **Lemma 5.** *For all $\mathbf{p}$ and all $\mathbf{q} \in \mathcal{C}(\mathbf{p}, \alpha_1, \alpha_2)$ with $\alpha_1 \leq 1/2$, for all $x$, we have $q_x \geq \frac{p_x}{N(1-\alpha_1)} \geq$*
640 *$\frac{p_x}{2N}$.*

641 *Proof.* Given $x$, let $B = \{x' : q_{x'} = q_x\}$. By calibration,

$$q_x = \frac{\mathbf{q}(B)}{|B|} \geq \frac{\mathbf{q}(B)}{N} \geq \frac{(1-\alpha_1)\mathbf{p}(B)}{N} \geq \frac{(1-\alpha_1)p_x}{N}.$$

642 If $\alpha_1 \leq 1/2$, we get $q_x \geq \frac{p_x}{2N}$. □

643 *Proof of Theorem 7.* By Lemma 5, we have $q_x \geq \frac{c_2 p_x}{N}$ for all $x$ with $c_2 = 0.5$. We apply Proposition
644 1, with all parameters exactly as in Theorem 3 except with $c_2 = 0.5$ rather than 1. □

Note that Theorem 7 is essentially identical to Theorem 3, up to a constant factor in $\beta$. Thus, all of our concentration results hold, up to constant factors, when $\mathbf{q} \in \mathcal{C}(\mathbf{p}, \alpha_1, \alpha_2)$ for $\alpha_1 \leq 1/2$ and *any* $\alpha_2$. Also note that Theorem 7 gives a high probability bound for any $\mathbf{q} \in C(\mathbf{p})$. If for example, we wish to minimize $\ell(\mathbf{q}; \mathbf{p})$ over some set of candidate calibrated distributions, we could form an $\epsilon$-net over these distributions and apply the theorem to all elements of this net, union bounding to obtain a bound on the probability that the empirical loss is close to the true loss on all elements. Optimizing would then yield a distribution with loss within $\gamma$ of the minimal.

## G  Details on Motivating Example

We now give details on the motivating example for considering alternatives to the log loss in the introduction (see Figure 1.)

**Dataset:**  Our primary data set is a list of 36663 of the most frequent English words, along with their frequencies in a count of all books on Project Gutenberg [3]. We then obtained a list of the 10000 most frequent French [1] and German [2] words. All capitals were converted to lower case, all accents removed, and all duplicates from the French and German lists removed. After preprocessing, the data consisted of the original 36663 English words along with 16409 French/German words. We gave the French and German words uniform frequency values, with the total frequency of these words comprising 12.23% of the probability mass of the word distribution.

Our tests are relatively insensitive to the exact frequency chosen for the French/German words within the reasonable range of 5-30%. Low frequency ($< 5\%$ of the total probability mass) is not sufficient noise to make the log loss minimizing distribution to perform poorly. On the other hand, high frequency ($> 30\%$ of the total probability mass) is too large and forces even our loglog loss minimizing distribution to perform poorly –due to its poor performance on the French and German words.

**Learning $\mathbf{q}_1$ and $\mathbf{q}_2$:**  We trained the candidate distribution $\mathbf{q}_1$ by minimizing log loss for a basic character trigram model. Minimizing log loss here simply corresponds to setting the trigram probabilities to their relative frequencies in the dataset. These frequencies were computed via a scan over all words in the dataset, taking into account the word frequencies. Note that we have full access to the target $\mathbf{p}$ and thus $\mathbf{q}_1$ exactly minimizes $\ell(\mathbf{q}; \mathbf{p}) = \mathbb{E}_{x \sim \mathbf{p}} \left[ \ln \frac{1}{q_x} \right]$ over all trigram models.

We trained $\mathbf{q}_2$ by distorting the optimization to place higher weight on the head of the distribution. In particular, we let $\bar{\mathbf{p}}$ be the distribution with $\bar{p}_x \propto p_x^\alpha$ for $\alpha = 1.4$. and minimized log loss over $\bar{\mathbf{p}}$. We saw similar performance for $\alpha \in [1.3, 2]$. Below this range, there was not significant difference between $\mathbf{q}_1$ and $\mathbf{q}_2$. Above this range, $\mathbf{q}_2$ placed very large mass on the head of the distribution, e.g., outputting the most common word `the` with probability $\geq .40$.

**Results:**  Our results are summarized in Figure 1. We can see that $\mathbf{q}_2$ seems to give more natural word samples and, while it achieves worse log loss than $\mathbf{q}_1$ (it must since $\mathbf{q}_1$ minimizes this loss over all trigram models), it achieves better log log loss. This indicates that in this setting, the log log loss may be a more appropriate measure to optimize. Our approach to training $\mathbf{q}_2$ via a reweighting of $\mathbf{p}$ can be viewed a heuristic for minimizing log log loss. Developing better algorithms for doing this, especially under the constraint that $\mathbf{q}_2$ is (approximately) calibrated is an interesting direction.

One way to see the improved performance of $\mathbf{q}_2$ is that its cumulative distribution more closely matches that of $\mathbf{p}$. See plot in Figure 1. Overall $\mathbf{p}$ places $87.77\%$ of its mass on the English words in the input distribution. $\mathbf{q}_1$ places $45.56\%$ of its mass on these words and $\mathbf{q}_2$ places $83.40\%$ of its mass on them. Note that the cumulative distribution plot and these statistics are *deterministic*, since $\mathbf{q}_1$ and $\mathbf{q}_2$ are trained by exactly minimizing log loss over the distributions $\mathbf{p}$ and $\bar{\mathbf{p}}$ without sampling. Thus no error bars are shown.

Below we show an extended sampling of words from $\mathbf{q}_1$, $\mathbf{q}_2$ and $\mathbf{p}$, evidencing $\mathbf{q}_2's$ superior performance on the task of generating natural English words. In this single run, e.g., $\mathbf{q}_1$ generates 6 distinct commonly used English words $\{$and, the, why, soon, caps, of$\}$. $\mathbf{q}_2$ generates 10: $\{$all, the, which, on, take, and, be, in, of, he$\}$. $\mathbf{p}$ generates 19, all with the except of the German word verweigert. More quantitatively, in a run of 10000 random samples, $\mathbf{p}$ generates 2497 distinct English words (the word distribution is very skewed so many duplicates of common words are generated). In comparison, $\mathbf{q}_1$ generates 815 distinct words and $\mathbf{q}_2$ generates 957.

Of course there are many methods of evaluating the performance of $\mathbf{q}_1$ and $\mathbf{q}_2$, which generally will be application specific. Our experiments are designed to give just a simple example, motivating the idea that minimizing log loss may not always be the optimal choice, and, like in classification and regression, there is room for alternative loss functions to be considered.

| Samples from $\mathbf{q}_1$ | Samples from $\mathbf{q}_2$ | Samples from p |
|---|---|---|
| and | all | old |
| tiest | the | verweigert |
| rike | which | five |
| agal | nesell | common |
| the | on | ny |
| itunge | whostionespirs | significance |
| cand | the | friend |
| ho | take | i |
| aren | the | with |
| why | and | museum |
| soon | be | the |
| ca | frould | without |
| caps | in | in |
| der | the | ethan |
| connestand | the | pointed |
| of | goich | def |
| per | of | down |
| shicy | ithe | the |
| theared | he | sky |
| introt | ong | the |

# H    Calibration Definition

In this section we give further discussion on our definition of calibration. Most typically in forecasting, calibration is a property of a *sequence* of forecasts $\mathbf{q}^{(1)}, \dots$, evaluated against a *sequence* of samples $x^{(1)}, \dots$. So our definition may require some background. First, we give a justification based on $\mathbf{q}$ as a coarsening of $\mathbf{p}$. Then, we show how formalizations of calibration for sequences of forecasts can be related to our definition.

**As a coarsening.**    One way to view the forecast $\mathbf{q}$ is as a coarsening of $\mathbf{p}$ in the sense of assigning probabilities to certain events $B_\alpha \subseteq \mathcal{X}$, but remaining agnostic as to the relative probabilities of various elements of $B_\alpha$, assigning all of them equal weight $\alpha$. By dividing $\mathcal{X}$ into maximal pieces $B_\alpha$ on which $\mathbf{q}$ is piecewise uniform, in this way one obtains that $\mathbf{q}$ is literally a coarsening of $\mathbf{p}$ if $\mathbf{p}(B_\alpha) = \mathbf{q}(B_\alpha)$ for each piece (as the pieces partition $\mathcal{X}$). This is our definition of calibration.

This directly captures the typical informal definition of calibration as "events that are assigned probability $\beta$ occur a $\beta$-fraction of the time", where the pieces $B_\alpha$ are the events and $\beta = \mathbf{q}(B_\alpha) = \mathbf{p}(B_\alpha)$ are the probabilities assigned to them.

It is also consistent with standard formalizations of calibration for sequences (see below), as if $x^{(s)} \sim \mathbf{p}$ i.i.d. each round and $\mathbf{q}^{(s)} = \mathbf{q}$ each round, one has that in the limit, each piece $B_\alpha$ will be represented as often as $\mathbf{q}$ predicts.

**Sequences of forecasts.**    Calibration of sequences can be formalized, for example, as follows. If each $x^{(t)} \in \mathcal{X} = \{0, 1\}$, then we can let $R_t$ be the set of rounds $s \le t$ where $x^{(s)} = 1$ and $S_t(\mathbf{q})$ be the set of rounds $s \le t$ where $\mathbf{q}^{(s)} = \mathbf{q}$. In this case, the sequence is termed *calibrated* if, on rounds where $\mathbf{q}$ was predicted, the fraction of times that $x^{(s)} = 1$ converges to $q_1$:

$$\forall \mathbf{q}: \qquad \lim_{t \to \infty} \frac{|S_t(\mathbf{q}) \cap R_t|}{|S_t(\mathbf{q})|} = q_1.$$

One way to obtain our definition is by "flattening" this one: let there be a finite number of rounds and suppose $\mathbf{p}, \mathbf{q}$ are probability distributions over rounds (so $\mathbf{p}$ will pick exactly one round to occur, and $\mathbf{q}$ assigns a binary prediction to each round). In this case we can let $S(\alpha) = \{t : q_t = \alpha\}$ be

the set of rounds assigned a probability $\alpha$ by the forecast, then naturally the round $t \sim \mathbf{p}$ lies in this set with probability $\mathbf{p}(S(\alpha))$. So the flattened definition of calibration requires that for each $\alpha$, $\mathbf{p}(S(\alpha)) = \mathbf{q}(S(\alpha))$, which is exactly our definition.

Our definition can also be obtained as described above by letting $\mathcal{X}$ be general, letting $\mathbf{q}$ be forecast on each round while $x^{(s)} \sim \mathbf{p}$ i.i.d. each round. If one interprets $\mathbf{q}$ as a distribution over events $B_\alpha$ that partition $\mathcal{X}$, one obtains the requirement that in the limit $\mathbf{p}(B_\alpha) = \mathbf{q}(B_\alpha)$ for each $\alpha$.

## I  Strong Properness in $\ell_2$ Norm

Our criteria can be extended to utilize different distance measures than our choice of $\ell_1$ or total variation distance. However, justifying and investigating other measures requires further work. In particular, this section shows why a choice of $\ell_2$ distance can be problematic.

Following our main definitions, one can define a loss to be strongly proper in $\ell_2$ if, for all $\mathbf{p}, \mathbf{q}$,

$$\ell(\mathbf{q}; \mathbf{p}) - \ell(\mathbf{p}; \mathbf{p}) \geq \frac{1}{2} \|\mathbf{p} - \mathbf{q}\|_2^2.$$

In particular, consider the quadratic loss $\ell(\mathbf{q}, x) = \frac{1}{2} \|\boldsymbol{\delta}^x - \mathbf{q}\|_2^2$, which can be shown to be 1-strongly-proper in $\ell_2$ (Corollary 3). However, the usefulness of this guarantee can be limited, as the following example shows.

**Proposition 2.** *Given a* 1-*strongly proper loss in $\ell_2$ norm, $\mathbf{q}$ can assign probability zero to the entire support of $\mathbf{p}$, yet have expected loss within $\frac{2}{N}$ of optimal.*

*Proof.* Let $\mathcal{X} = \{1, \ldots, N\}$ for even $N$. Let $\mathbf{p}$ be uniform on $\{1, \ldots, \frac{N}{2}\}$ and let $\mathbf{q}$ be uniform on $\{\frac{N}{2} + 1, \ldots, N\}$.

The point is that for any such "thin" distributions (small maximum probability), their $\ell_2$ norms $\|\mathbf{p}\|, \|\mathbf{q}\|$ are vanishing and by the triangle inequality so is the distance $\|\mathbf{p} - \mathbf{q}\|$ between them.

In this example, $\|\mathbf{p} - \mathbf{q}\|_2^2 = N \left(\frac{2}{N}\right)^2 = \frac{4}{N}$. So strong properness only guarantees that the difference in loss is $\ell(\mathbf{q}; \mathbf{p}) - \ell(\mathbf{p}; \mathbf{p}) \geq \frac{2}{N}$. In fact, this is exactly matched by the quadratic loss, where the difference in expected score (the Bregman divergence of the two-norm) is exactly $\frac{1}{2} \|\mathbf{p} - \mathbf{q}\|_2^2 = \frac{2}{N}$. $\qquad\square$

Thus, strongly proper losses in $\ell_2$ can converge to optimal expected loss at the rapid rate of $O(\frac{1}{N})$ even when making completely incorrect predictions.

## J  Strongly Proper Losses and Scoring Rules on the Full Domain

In this section, for completeness, we investigate the strongly proper criterion in the traditional setting of proper losses (equivalently, scoring rules). The main result is that, just as (strictly) proper losses are Bregman divergences of (strictly) convex functions, so are *strongly* proper losses Bregman divergences of *strongly* convex functions. We derive some non-local strongly proper losses. These results may be of independent interest.

**Terminology.**  Given a function $f : \mathbb{R}^d \to \mathbb{R}$, the vector $v \in \mathbb{R}^d$ is a *supergradient* of $f$ at $z$ if for all $z'$, we have $f(z') \leq f(z) + v \cdot (z' - z)$. (In other words, there is a tangent hyperplane lying above $f$ at $z$ with slope $v$.) A function is *concave* if it has at least one supergradient at every point. (If exactly one, it is differentiable.) In this case, use $df(z)$ to denote a choice of a supergradient of $f$ at $z$.

Given a concave $f$, the *divergence function of $f$* is

$$D_{-f}(z, z') := [f(z') + df(z') \cdot (z - z')] - f(z),$$

the gap between $f(z)$ and the linear approximation of $f$ at $z'$ evaluated at $z$. The reason for this notation is that $D_{-f}$ is the Bregman divergence of the convex function $-f$.

**Definition 9** (Strongly Concave). A function $f : \mathbb{R}^d \to \mathbb{R}$ is $\beta$-*strongly concave* with respect to a norm $\|\cdot\|$ if for all $z, z'$,

$$D_{-f}(z, z') \geq \frac{\beta}{2} \|z - z'\|^2.$$

## J.1 Background: proper loss characterization

We first recall some background from theory of proper scoring rules, phrased in the loss setting. Given a loss $\ell(\mathbf{q}, x)$, the expected loss function is $H_\ell(\mathbf{p}) = \ell(\mathbf{p}; \mathbf{p})$. The following classic characterization says that (strict) properness of $\ell$ is equivalent to (strict) concavity of $H_\ell$.

**Theorem 8** ([21, 24, 15])**.** $\ell$ is a (strictly) proper loss if and only if $H_\ell$ is (strictly) concave. If so, we must have
$$\ell(\mathbf{q}, x) = H_\ell(\mathbf{q}) + dH_\ell(\mathbf{q}) \cdot (\boldsymbol{\delta}^x - \mathbf{q})$$
where $dH_\ell(\mathbf{q})$ is any supergradient of $H_\ell$ at $\mathbf{q}$ and $\boldsymbol{\delta}^x$ is the point mass distribution on $x$.

**Corollary 1.** The expected loss of $\mathbf{q}$ under true distribution $\mathbf{p}$ is the linear approximation of $H_\ell$ at $\mathbf{q}$, evaluated at $\mathbf{p}$:
$$\ell(\mathbf{q}; \mathbf{p}) = H_\ell(\mathbf{q}) + dH_\ell(\mathbf{q}) \cdot (\mathbf{p} - \mathbf{q}).$$

**Corollary 2.** When the true distribution is $p$, the improvement in expected loss for reporting $\mathbf{p}$ instead of $\mathbf{q}$ is the divergence function of $H_\ell$ (the Bregman divergence of $-H_\ell$), i.e.
$$\ell(\mathbf{q}; \mathbf{p}) - \ell(\mathbf{p}; \mathbf{p}) = D_{-H_\ell}(\mathbf{p}, \mathbf{q}).$$

**Example 5.** Recall from Example 1 the log loss $\ell(\mathbf{q}, x) = \ln \frac{1}{\mathbf{q}}$ has expected loss equal to Shannon entropy. The associated Bregman divergence is the KL-divergence, so the difference in expected log loss between $\mathbf{p}$ and $\mathbf{q}$ under true distribution $\mathbf{p}$ is $KL(\mathbf{p}, \mathbf{q}) := \sum_x p_x \ln \frac{p_x}{q_x}$. The quadratic loss has expected loss $H_{\text{quad}}(\mathbf{p}) = \frac{1}{2} - \frac{1}{2}\|\mathbf{p}\|_2^2$, so the associated Bregman divergence is $D_{-H_{\text{quad}}}(\mathbf{p}, \mathbf{q}) = \frac{1}{2}\|\mathbf{p} - \mathbf{q}\|_2^2$.

The above are all well-known, although in the literature on proper scoring rules everything is negated (a score is used equal to negative loss, the expected score is convex, etc.).

## J.2 Strongly concave functions and strong properness

Given the above characterization and our (carefully chosen) definition of *strongly proper*, the classic characterization of proper losses extends easily:

**Theorem 9.** A proper loss function $\ell$ is $\beta$-strongly proper (with respect to a norm) if and only if $H_\ell$ is $\beta$-strongly concave (with respect to that norm).

*Proof.* We have $\ell(\mathbf{q}; \mathbf{p}) - \ell(\mathbf{p}; \mathbf{p}) = D_{-H_\ell}(\mathbf{p}, \mathbf{q})$ by Corollary 2. $H_\ell$ is $\beta$-strongly concave if and only if $D_{-H_\ell}(\mathbf{p}, \mathbf{q}) \geq \frac{\beta}{2}\|\mathbf{p} - \mathbf{q}\|$ for all $\mathbf{p}, \mathbf{q}$, which is the condition that $\ell$ is $\beta$-strongly proper. $\quad\square$

Though the proof is trivial once the definitions are set up and followed through, the statement is powerful. It completely characterizes the proper loss functions satisfying that, if $\mathbf{q}$ is significantly wrong (far from $\mathbf{p}$), then its expected loss is significantly worse. It also gives an immediate recipe for constructing such losses: Start with any concave function $H(\mathbf{q})$ that is strongly concave in your norm of choice, and set $\ell(\mathbf{q}, x) = H(\mathbf{q}) + dH(\mathbf{q}) \cdot (\boldsymbol{\delta}^x - \mathbf{q})$. All strongly proper losses satisfy this construction for some such $H$.

## J.3 Known examples

Recall that the log scoring rule's expected loss function is Shannon entropy. Hence, the fact that log loss is 1-strongly-proper (Example 2) turns out to be equivalent to the statement that Shannon entropy is 1-strongly convex in $\ell_1$ norm. As described in Section 3, this fact (perhaps surprisingly) is equivalent to Pinsker's inequality.

However, $\ell_1$-strong properness seems difficult to satisfy over the simplex. In particular,

**Proposition 3.** The quadratic scoring rule is not strongly proper in $\ell_1$ norm.

*Proof.* Consider $\mathbf{q}$ as the uniform distribution and let $p_x \in \frac{1\pm\epsilon}{N}$, such that $\|\mathbf{p} - \mathbf{q}\|_1 = \epsilon$. Then $\ell(q; p) - \ell(p; p) = \frac{1}{2}\|\mathbf{p} - \mathbf{q}\|_2^2 = \frac{1}{2}(N)\left(\frac{\epsilon}{N}\right)^2 = \frac{\epsilon^2}{2N}$. As $N \to \infty$, this difference in loss goes to zero while $\|\mathbf{p} - \mathbf{q}\|_1 = \epsilon$, so there is no fixed $\beta$ such that the loss is $\beta$-strongly proper. $\quad\square$

808  We can show that it is strongly proper in $\ell_2$ norm. However, the usefulness of $\ell_2$ strong properness is
809  less clear, as is demonstrated in Appendix I.

810  **Lemma 6.** *The function* $-\frac{1}{2}\|\mathbf{p}\|_2^2$ *is* 1-*strongly concave with respect to the* $\ell_2$ *norm.*

811  *Proof.* The associated Bregman divergence is $\frac{1}{2}\|\mathbf{p} - \mathbf{q}\|_2^2$, which is equal to $\frac{1}{2}\|\mathbf{p} - \mathbf{q}\|_2^2$, so it is
812  1-strongly convex in $\ell_2$ norm. $\qquad\square$

813  **Corollary 3.** *The quadratic loss is* 1-*strongly proper with respect to the* $\ell_2$ *norm.*

814  ## J.4  New proper losses

815  Because the $\ell_1$ norm is especially preferred when measuring distances between probability distribu-
816  tions, we seek losses that are 1-strongly proper with respect to the $L_1$ norm. By the characterization of
817  Theorem 9, this is equivalent to seeking $\ell_1$ $\beta$-strongly-convex functions of probability distributions.

818  **Lemma 7.** *Let* $M \in \mathbb{R}^{N \times N}$ *be the* negative *of the Hessian of a function* $H : \Delta_{\mathcal{X}} \to \mathbb{R}$. *Then* $H$ *is*
819  $\beta$-*strongly concave in* $\ell_1$ *norm if and only if*

$$\min_{w : \|w\|_1 = 1} w^\intercal M w \geq \beta$$

820  *Proof.* $M$ is the Hessian of the (presumably convex) function $-H$. $\qquad\square$

821  We focus on *separable, symmetric* concave functions: $H(\mathbf{q}) = \sum_x h(q_x)$ for some concave function
822  $h$. In this case the Hessian of $H$ is a diagonal matrix with $(x, x)$ entry $\frac{d^2 h(z)}{dz^2}$. Call its negative $M$ as
823  in Lemma 7 and for convenience later, let us define $f(z)$ as

$$\frac{1}{f(z)} := \frac{-d^2 h(z)}{dz^2}.$$

824  Then by Lemma 7, $H(\mathbf{q})$ is $\beta$-strongly concave if

$$\beta \leq \min_{w : \|w\|_1 = 1} w^\intercal M w$$

$$= \min_{w : \|w\|_1 = 1} \sum_x \frac{w_x^2}{f(q_x)}.$$

825  This is solved by setting $w_x \propto f(q_x)$, where the normalizing constant is $C := \sum_x f(q_x)$. So we have

$$\beta \leq \sum_x \left( \frac{f(q_x)}{C} \right)^2 \frac{1}{f(q_x)}$$

$$= \frac{1}{C^2} \sum_x f(q_x)$$

$$= \frac{1}{C}$$

$$= \frac{1}{\sum_x f(q_x)}.$$

826  So for 1-strong concavity, we require $\sum_x f(q_x) \leq 1$ for all $\mathbf{q}$. Now choose $f(q_x) = q_x^{1+\alpha}$.

827  - If $\alpha < 0$, then $\sum_x f(q_x)$ can be arbitrarily large and the resulting function is not strongly
828    concave in $\ell_1$ norm.
829  - If $\alpha = 0$, then we have $\frac{d^2 h(z)}{dz^2} = \frac{-1}{z}$ and we recover $h(z) = z \ln(\frac{1}{z})$, which gives $H$ as
830    Shannon entropy; the log scoring rule.
831  - If $\alpha \geq 1$, we get $h(z)$ is unbounded on $[0, 1]$, so we obtain an expected loss function that is
832    unbounded on the simplex.
833  - For $0 < \alpha < 1$, we get a class of apparently-new proper loss functions that are 1-strongly
834    proper. Here $\frac{d^2 h(z)}{dz^2} = \frac{-1}{z^{1+\alpha}}$, so $h(z) = z^{1-\alpha}$ and $H(z) = \sum_x q_x^{1-\alpha}$.

835  In particular, for the last class, we identify the appealing case $\alpha = 0.5$. It gives the following "inverse
836  root" loss function:

837    - $H(\mathbf{q}) = 2 \sum_x \sqrt{q_x}$.
838    - $\ell(\mathbf{q}, x) = \frac{1}{\sqrt{q_x}} + \sum_{x'} \sqrt{q_{x'}}$.
839    - $\ell(\mathbf{q}; \mathbf{p}) = \sum_x \frac{1}{\sqrt{q_x}} (p_x + q_x)$.
840    - $D_{-H}(\mathbf{p}, \mathbf{q}) = \sum_x \frac{1}{\sqrt{q_x}} \left( \sqrt{p_x} - \sqrt{q_x} \right)^2$.

841    We are not aware of this loss having been used before, but it seems to have nice properties. There is
842    an apparent similarity to the squared Hellinger distance $H(\mathbf{p}, \mathbf{q})^2 := \frac{1}{2} \sum_x \left( \sqrt{p_x} - \sqrt{q_x} \right)^2$, we are
843    not aware of a closer formal connection. For example, Hellinger distance is symmeteric.