[Reviews · NeurIPS 2019]

Reviewer 1



The paper studied loss functions for distribution approximation. It define various properties and prove different Theorems that show some nice inequalities with respect to these properties. Yet, the main punchline in the paper is missing. Why these properties are really important? I would expect to see a real example where the analysis made in the paper helps with. This part is lacking in the paper. So it is not really clear what is the applicability of the derived results. Also the claim that only log-loss is used for learning distributions is not true. Unless I miss something, generative models are designed to learn distributions and they don't use only the log-loss. Where do they fit in the story? I think that giving them as an example and studying for example the earth moving distance and seeing what properties it gives etc. will make the paper stronger and clearer of why it is important. At the moment, it is far from clear to me.

Reviewer 2



The paper is written on a good level and includes the ideas which allow to characterize the properties of loss-function for distribution learning on the finite discrete domain. Main such property is the calibration which allows to state several results on the size of sample size which ensures concentration of the empirical loss around true loss. However, there are some points, which to my mind should have been enlighted more in paper. Namely, for the case of log-loss function optimizing a loss-function with respect to measure $q$ is the same as minimizing the KL-divergence for given measure p and q - to choose. Authors only firstly mention this when considering the example of properness of the log-loss function. Results for the concentration sample size of the empirical loss function (provided restriction to calibrated distributions) are interesting, but, taking into account the folklore result on the log-loss sample complexity are not surprising. Several typos: line 137: will be are characterized line 138 : we will generally assume functions are differentiable. Efficient implementation strategy would be interesting to consider in the paper as well as the explicit procedure how we build the estimator $\hat{p}$. Is it the estimator based on the relative frequencies of occuring of each element? In this case what value should be considered for the elements which are in the domain but did not occure in the sample?

Reviewer 3



Originality: This work is original; the idea of restricting to calibrated distributions and subsequently finding a condition to obtain properness is new AFAIK. Clarity: For readers somewhat familiar with the material surrounding the paper content, the paper is clear. The organization of the paper is good. All of the proofs I checked were correct, aside from typos/what I think are minor errors that do not affect the result. However, I think the example from lines 218 to 223 is incorrect, where the authors try to show an example where the linear loss is not calibrated sample proper. Significance: For me, this is the weak point of the paper. Firstly, I think Theorems 3,4 can be improved (see comments in contributions section). Second, though mathematically nice, the restriction to calibrated distributions seems potentially problematic. Knowledge of the set of calibrated distributions is essentially knowledge of the target distribution itself. Thus, it is unclear why properness under restriction to calibrated distributions is significant. However, there may be a way to simultaneously learn calibrated distributions and minimize a loss over currently learned calibrated distributions; if the authors show something like that, in my opinion it would be very significant (although that seems worthy of another paper). The note on constructing approximately calibrated distributions in the appendix is a step towards this, but is far from it. Quality: I would score 6 or 7 at the moment. Here is a list of errata/points to be clarified: - Line 137: "Our main results will be are" - Line 214: "Since it is not a proper loss function over \Delta_X, by definition it is not sample proper over \Delta_X for any m(\epsilon, \delta, N)." I don't follow this justification - Line 221: If q_x is equal to 0 for x=N/2 + 1,..., N, how is it calibrated w.r.t p_x, which is nonzero for x=N/2+1,...,N? Also, \sqrt{m} \pm 2\sqrt{m} seems off, and I do not know why there is a -\Theta(1/N). - Line 237: reasonable loss -> reasonable losses - Theorem 1 should probably say strictly concave function f - Line 301: for where -> where - Line 412: Equation underneath should have a minus sign instead of a plus sign - Derivation under line 438: I think the third line should have an 8 in the denominator, not a 4. Also, I don't think \epsilon \leq p(B) is generally true, since if p=[0,0,0,0,1/2,1/2], B={1,2,3,4,5,6}, then t(B) = 1/6, so \epsilon = \sum_i |p_i-1/6| = 4/3, but p(B) = 1. I think \epsilon \leq 2p(B) is true, which adds another factor of 2 to the denominator. With the previous change, there is a 16 in the denominator, which doesn't change the end result. - Line 457: parenthesis in subscript - Line 463: I think c_2^{1/(1-r)} should be in the denominator, not c_2^{r/(1-r)} - Line 466: What's the point of Bernstein's inequality here instead of the simpler Hoeffding's? - Line 474 derivation: Why is \sum_x p_x^{1-r} = N^r? (It is \leq N, which still lets the result go through) - Line 477 equation underneath: How do you get a net power of N^{r-2}? I get N^{2r-3} (which still lets the result go through) - Observation 1. in appendix E does not seem to be correct, as despite e^{-x} being nonnegative, twice differentiable, decreasing, and convex, e^{-1/x} is not concave on strictly positive reals. - For the proof of observation 2, casework seems unnecessary, and there is an extra minus sign under line 509. - Line 537: q_t should be q_x. Edit after response: I think the authors' fixed example works. Now the linear loss is not calibrated sample proper when m is o(N), which is still saying something since the authors give losses that are calibrated sample proper with m=O(polylog(ln N)). I buy that with constant probability, \hat p_1 \leq 1/4 - \Theta(1/sqrt{m}) and \hat p_2 \leq 1+ 1/4 + \Theta(1/\sqrt{m}), since with constant probability a pair of i.i.d. Gaussians are less than 1 sd below/greater than 1 sd above their mean, but a more rigorous argument should be used in the final version. On the calibration assumption, I think the authors clarifying that this work is currently more applicable to evaluating algorithm outputs than designing learning algorithms, and the references cited on designing calibrated prediction schemes are good justification for the work. It would still be nice to have a clearer link between this work and existing distribution learning algorithms. My final score is a strong 7.

[Author Response · NeurIPS 2019]

We thank the reviewers for their feedback. Below we address specific comments.

**All reviewers**: We emphasize that our paper examines ways to *evaluate* distributions produced by *any learning*
*algorithm*. While it can be used for designing specific learning algorithms in future, that is not our primary objective.

**Reviewer #2:** The reviewer asks for a clarification of the main message. We answer this question in 3 parts.

• **Importance of the properties we explored:** The importance of properness has been well established in the literature
and is a major motivation behind the popularity of log loss minimization. As we mentioned, if the loss is not proper
then even if the learner receives infinitely many samples the loss minimization recovers a wrong distribution. Sample
properness is a natural extension of properness to finite samples. If a loss function is not sample-proper, then loss
minimization on no reasonable amount of samples would lead to the right distribution. As we mentioned, sample
properness is essential for distribution learning and has been studied in previous work for the specific case of log-loss.
Concentration is also a natural property of losses, without it the estimated loss of a candidate distribution from
samples is not a good proxy for its actual loss, and therefore, cannot be used for the purpose of distribution evaluation.
• **Prevalence of loss minimization for distribution learning:** A generative model is one that generates samples from
some underlying distribution. Thus in our context, a generative model is exactly a candidate distribution $\mathbf{q}$ which
is meant to estimate the target distribution $\mathbf{p}$. There are many methods to train and evaluate generative models,
the most popular of which is the average log likelihood that the model assigns to a sample set. For example, in
GANs (Goodfellow et al. 2014) the focus is on average likelihood maximization, *which is equivalent to log loss*
*minimization.* Some of the other methods such as Jensen-Shannon divergence, contrastive divergence, etc. are also
closely related to log loss minimization.
• **Clarification of main contributions:** The reviewer mentions that we study log-loss with respect to the above
properties. While we do mention log-loss in this way, our main contribution is to show that many functions in
addition to log-loss meet these properties when we consider calibrated distributions. On the other hand, no function
(not even log-loss) possess these properties without calibration. As we showed in Figure 1, in some applications
log loss is not the optimal loss function to be used. By putting forward alternative loss functions with desirable
properties, our work paves the way for picking loss functions based on the domain's need.

**Reviewer #3:**

• **Concentration Results:** We disagree that our concentration results are not surprising given the folklore theorem. As
we mention in the paper even the log loss (while being sample proper by the folklore theorem) *does not concentrate*
without a calibration assumption. That is, our concentration result indeed has to leverage the structural properties of
calibration in addition to the inverse concavity of the loss.
• **Definition of $\hat{\mathbf{p}}$:** As mentioned $\hat{\mathbf{p}}$ denotes the empirical distribution. As is standard, the probability of an element in
$\hat{\mathbf{p}}$ is its relative frequency in the sample. Elements in the domain and not in the sample are assigned probability 0.
• **Efficient implementation:** We agree that efficient algorithms are an important direction for future work. We included
some preliminary results in the supplemental (see Section 5) in this regard. Our main goal in this paper however is to
lay out a formal foundation for evaluating distributions via desirable loss functions that future algorithms can rely on.

**Reviewer #4:** Thank you for your helpful and detailed comments, which we will implement in the final version.

• **Example**: We agree with this and have corrected the example. At a high level elements $3, \ldots, N$ of $\mathbf{q}$ and $\mathbf{p}$ should
have been switched, i.e., $q_{3:N} = 1/2(N-2)$. This fixes the calibration issue. As for sample-properness, note that
the contribution of elements $x = 3 \ldots, N$ is at most $\Theta(1/N)$ to the equation in line 218. With a constant probability
$\hat{p}_1 \leq \frac{1}{4} - \frac{1}{\sqrt{m}}$ and $\hat{p}_2 \geq \frac{1}{4} + \frac{1}{\sqrt{m}}$. That is, $\ell(\mathbf{q}; \hat{\mathbf{p}}) - \ell(\mathbf{p}; \hat{\mathbf{p}}) \leq -1/m + \Theta(1/N) < 0$ for $m \in o(N)$. It is possible
to strengthen this bound with a more careful analysis of the contribution of elements $3, \ldots, N$, but the main message
of this part remains the same regardless, that is, *linear-loss is not sample-proper.*
• **Is calibration a natural assumption:** Calibration has been used in a long line of work (including [11, 13]) as a
natural requirement for probabilistic predictions. We consider a major contribution of our work to be in understanding
how the classic notion of calibration relates to loss functions for evaluating/learning distributions, and in particular in
showing that this restriction circumvents the impossibility result for satisfying the three natural criteria we considered.
• **On"for all distributions" results:** We agree that this is an interesting direction. One could strengthen Thms. 3 & 4
by taking a union bound over a finite net over all distributions. Thus, our results directly apply to this setting albeit
with worsened sample complexity. We note that this worsening of sample complexity is needed, because the "for all
distribution" version of our result would enable distribution learning in the worst-case, which is known to require
$\Omega(N)$ samples. We note that there are methods that can perform ERM over a full class of distributions when the
target distribution is not a worst-case distribution. Many such methods (e.g., GANs) perform an implicit optimization,
and then are evaluated using various losses. In this case, the guarantees given in Theorem 3 & 4 are directly useful
– they show that a small number of samples suffices to compare a finite set of outputs from various algorithms or
parameter settings accurately.

[Meta-Review · NeurIPS 2019]

The manuscript analyzes various properties of loss functions for probabilistic estimation, and provides a number of results on properness and calibration. As pointed out by the reviewers, the strengths of the manuscript include the novelty and timeliness of the problem of interest, clarity of writing, and novelty of the results. The reviewers also point out some weaknesses, included limited generality of the the main results -- particularly related to properness. There were also very limited empirical motivation. The primary empirical result on outliers behavior of log losses for language tasks was unclear -- if this was an complete experiment by the authors, and if so why a more thorough evaluation was not provided. Overall, the reviewers and AC agree that this is an interesting submission, that has some potential to encourage new theoretical and applied work on loss functions, and thus may be of general interest.